# Learning Fast, Learning Slow: A General Continual Learning Method based on Complementary Learning System

**Elahe Arani**\*, **Fahad Sarfraz**\* **& Bahram Zonooz**
Advanced Research Lab, NavInfo Europe, Eindhoven, Netherlands
{elahe.arani, fahad.sarfraz}@navinfo.eu, bahram.zonooz@gmail.com

## Abstract

Humans excel at continually learning from an ever-changing environment whereas it remains a challenge for deep neural networks which exhibit catastrophic forgetting. The complementary learning system (CLS) theory suggests that the interplay between rapid instance-based learning and slow structured learning in the brain is crucial for accumulating and retaining knowledge. Here, we propose CLS-ER, a novel dual memory experience replay (ER) method which maintains short-term and long-term semantic memories that interact with the episodic memory. Our method employs an effective replay mechanism whereby new knowledge is acquired while aligning the decision boundaries with the semantic memories. CLS-ER does not utilize the task boundaries or make any assumption about the distribution of the data which makes it versatile and suited for "general continual learning". Our approach achieves state-of-the-art performance on standard benchmarks as well as more realistic general continual learning settings. [1]

## 1 Introduction

Continual learning (CL) refers to the ability of a learning agent to continuously interact with a dynamic environment and process a stream of information to acquire new knowledge while consolidating and retaining previously obtained knowledge (Parisi et al., 2019). This ability to continuously learn from a changing environment is a hallmark of intelligence and a critical missing component in our quest towards making our models truly intelligent. The major challenge towards enabling CL in deep neural networks (DNNs) is that the continual acquisition of incrementally available information from non-stationary data distributions leads to catastrophic forgetting whereby the performance of the model on previously learned tasks drops drastically (McCloskey & Cohen, 1989).

Several approaches have been proposed to address the issue of catastrophic forgetting in CL. These can be broadly categorized into regularization-based methods (Farajtabar et al., 2020; Kirkpatrick et al., 2017; Ritter et al., 2018; Zenke et al., 2017) which penalizes changes in the network weights, network expansion-based methods (Rusu et al., 2016; Yoon et al., 2017) which dedicate a distinct set of network parameters to distinct tasks, and rehearsal-based methods (Chaudhry et al., 2018; Lopez-Paz & Ranzato, 2017) which maintains a memory buffer and replays samples from previous tasks. Amongst these, rehearsal-based methods have proven to be more effective in challenging CL tasks (Farquhar & Gal, 2018). However, an optimal approach for replaying memory samples and constraining the model update to efficiently consolidate knowledge remains an open question.

In the brain, the ability to continually acquire, consolidate, and transfer knowledge over time is mediated by a rich set of neurophysiological processing principles (Parisi et al., 2019; Zenke et al., 2017) and multiple memory systems (Hassabis et al., 2017). In particular, the CLS theory (Kumaran et al., 2016) posits that efficient learning requires two complementary learning systems: the hippocampus exhibits short-term adaptation and rapid learning of episodic information which is then gradually consolidated to the neocortex for slow learning of structured information. Furthermore, a recent study by Hayes et al. (2021) identified the missing elements of biological reply in the replay

---

\*Contributed equally.
[1]The code is avaiable at: https://github.com/NeurAI-Lab/CLS-ER

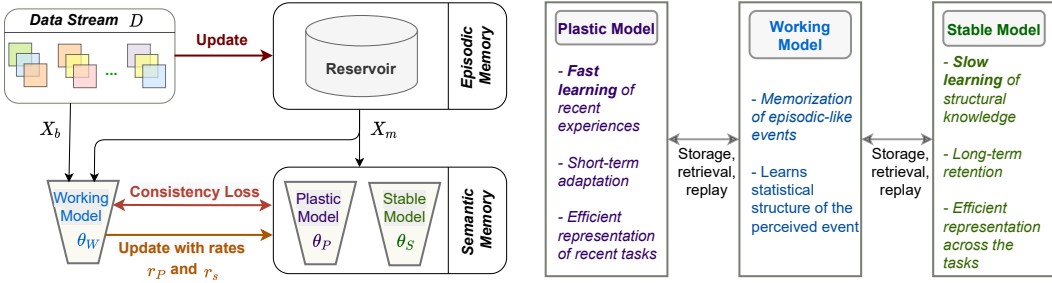

Figure 1: CLS-ER employs a dual-memory learning mechanism whereby the episodic memory stores the samples and the semantic memories build short-term and long-term memories of the learned representations of the working model. The two memories interact to enforce a consistency loss on the working model which prevents rapid changes in the parameter space and enables the alignment of the decision boundary with semantic memories for effective knowledge consolidation.

mechanisms employed in DNNs for CL. They highlight that many existing approaches only focus on modeling the prefrontal cortex directly and do not have a fast learning network which plays a critical role in enabling efficient CL in the brain. Inspired by these studies, we hypothesize that mimicking the slow and rapid adaptation of information and having an efficient mechanism for incorporating them into the working memory can enable better CL in DNNs.

To this end, we propose a novel dual memory experience replay method based on the complementary learning systems theory in the brain, dubbed as CLS-ER. In addition to a small episodic memory, our method builds long-term and short-term semantic memories which mimic the rapid and slow adaptation of information (Figure 1). As the network weights encode the learned representations of the tasks (Krishnan et al., 2019), the semantic memories are maintained by taking the exponential moving average of the working model's weights to consolidate information across the tasks with varying time windows and frequencies. The semantic memories interact with the episodic memory to extract consolidated replay activation patterns and enforce a consistency loss on the update of the working model so that new knowledge is acquired while aligning the decision boundary of the working model with the decision boundaries of semantic memories. This maintains a balance between the plasticity and stability of the model for effective knowledge consolidation.

CLS-ER provides a general CL method that does not utilize the task boundaries or make any strong assumption regarding the distribution of the data and tasks. We demonstrate the versatility and effectiveness of our method on a wide range of CL benchmark tasks as well as more challenging scenarios which simulate the complexities of CL in the real world.

## 2 RELATED WORK

The base method for the rehearsal-based approach, Experience Replay (ER) (Riemer et al., 2018) combines the memory samples with the task samples into the training batch. Several techniques have since been employed on top of ER. Meta Experience Replay (MER) (Riemer et al., 2018) considers replay as a meta-learning problem for maximizing the transfer from previous tasks and minimizing the interference. iCARL (Rebuffi et al., 2017) uses the nearest average representation of past exemplars to classify in an incrementally learned representation space. Gradient Episodic Memory (GEM) (Lopez-Paz & Ranzato, 2017) formulates optimization constraints on the exemplars in memory. Gradient-based Sample Selection (GSS) (Aljundi et al., 2019) aims for memory sample diversity in the gradient space and provides a greedy selection approach. Function Distance Regularization (FDR) (Benjamin et al., 2018) saves the network response at the task boundaries and adds a consistency loss on top of ER. Dark Experience Replay (DER++) applies knowledge distillation (Sarfraz et al., 2021) and regularization on logits sampled during the optimization trajectory.

CLS has been used as a source of inspiration for dual memory learning systems in earlier works (French, 1999; Robins, 1993) but they have not been shown to scale to current computer vision tasks (Parisi et al., 2019). Recently, Rostami et al. (2019) utilizes a generative model to couple sequential tasks in a latent embedding space. Kamra et al. (2017) utilizes two generative models

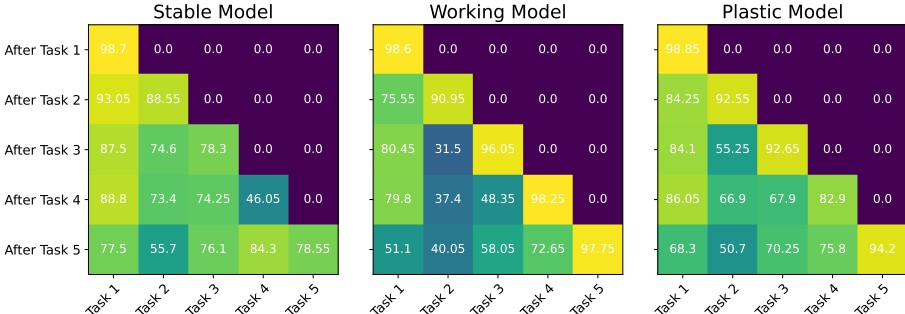

Figure 2: Task-wise performance on S-CIFAR-10 test set with 500 buffer size. The models are evaluated at the end of each task (y-axis) to evaluate how the task performances (x-axis) are affected as training progress. The stable model retains information from earlier tasks while the plastic model quickly adapts to the recent task. Note that there is less forgetting in the semantic memories compared to the working model. For other buffer sizes and S-TinyImageNet see Figures S1 and S2.

in a dual memory architecture. However, they utilize the task boundaries and generative replay has its own set of challenges as it is difficult to learn a faithful distribution and performs sub-par in comparison to instance-based replay methods on challenging CL settings. Generally, the inspiration from CLS theory in DNNs has been mostly limited to episodic memory and mimicking the rapid and slow learning mechanism is majorly ignored (Hayes et al., 2021) which we aim to address.

## 3 METHOD

We first provide an overview of the CLS theory for the brain and how we aim to mimic it for DNNs before introducing the main components of our method and the overall formulation.

### 3.1 COMPLEMENTARY LEARNING SYSTEM THEORY

The CLS theory posits that effective lifelong learning in the brain requires two complementary learning systems. The hippocampus rapidly encodes novel information as a short-term memory which is subsequently used to transfer and consolidate knowledge in the neocortex which gradually acquires structured knowledge representation as long-term memory through experience replay. The interplay between the functionality of the hippocampus and neocortex is crucial for concurrently learning efficient representations (for better generalization) and the specifics of instance-based episodic memory.

### 3.2 COMPLEMENTARY LEARNING SYSTEM BASED EXPERIENCED REPLAY

Inspired by the CLS theory, we propose a dual memory experience replay method, CLS-ER, which aims to mimic the interplay between fast learning and slow learning mechanisms for enabling effective CL in DNNs. Our method maintains short-term and long-term semantic memories of the encountered tasks which interact with the episodic memory for replaying the associated neural activities. The working model is updated so that it acquires new knowledge while aligning its decision boundary with the semantic memories to enable the consolidation of structured knowledge across the tasks. Figure 1 highlights the parallels between CLS theory and our method.

**Semantic Memories:** Central to our method is the maintenance of two semantic memories which accumulate and consolidate information over long-term and short-term periods. As the acquired knowledge of the learned tasks is encoded in the weights of DNNs (Krishnan et al., 2019), we aim to form our semantic memories by accumulating the knowledge encoded in the corresponding weights of the model as it sequentially learns different tasks.

An efficient method for aggregating the weights of a model is provided by *Mean Teacher* (Tarvainen & Valpola, 2017) which is a knowledge distillation approach that uses an exponential moving average (EMA) of the student's weights during training as a teacher for semi-supervised learning. It can also be considered as forming a self-ensemble of the intermediate model states that leads to better

internal representations. We adapt the Mean Teacher approach to build our semantic memories as it provides a computational and memory-efficient method for accumulating knowledge over the tasks.

As CL involves learning tasks sequentially, the model weights at each training step can be considered as a student model specialized for a particular task. Therefore, averaging the weights during training can be considered as forming an ensemble of task-specific student models which effectively aggregates information across the tasks and leads to smoother decision boundaries. CLS-ER builds long-term (stable model) and short-term (plastic model) semantic memories by maintaining two EMA-weighted models over the working model's weights. The stable model is updated less frequently with a larger window size so that it retains more information from the earlier tasks while the plastic model is updated more frequently with a smaller window size so that it adapts faster to information from new tasks (Figure 2). Section D further demonstrates the benefits of employing two semantic memories instead of a single semantic memory.

**Episodic Memory:** Replay of samples from the previous tasks stored in a small episodic memory is a common approach in CL that has proven to be effective in mitigating catastrophic forgetting. As we aim to position CLS-ER as a versatile general incremental learning method, we do not utilize the task boundaries or make any strong assumptions about the distribution of the tasks or samples. Therefore, to maintain a fixed episodic memory buffer, we employ *Reservoir sampling* (Vitter, 1985) which assigns equal probability to each sample in the stream for being represented in the buffer and randomly replaces the existing memory samples (Algorithm 2). It is a global distribution matching strategy that ensures that at any given time the distribution of samples in the buffer will approximately match the distribution of all the samples seen so far (Isele & Cosgun, 2018).

**Consolidation of Information:** The key challenge in CL is the consolidation of new information with the previously acquired information. This requires an effective balance between the stability and plasticity of the model. Furthermore, the sharp change in decision boundary as a new task is learned makes the consolidation of information over tasks more challenging. CLS-ER tackles these challenges through a novel dual memory experience replay mechanism. The long-term and short-term semantic memories interact with the episodic memory to extract the consolidated activations for the memory samples which are then utilized to constrain the update of the working model so that new knowledge is obtained whilst the decision boundary is aligned with the semantic memories. This prevents rapid changes in the parameter space as new tasks are learned. Furthermore, aligning the working model's decision boundary with the semantic memories serves two goals: (i) helps in retaining and consolidating information and (ii) leads to a smoother adaptation of decision boundary.

### 3.3 FORMULATION

CLS-ER involves training a working model $f(.; \theta_w)$ on a data stream $\mathcal{D}$ sampled from a non-iid distribution. Two additional EMA-weighted models are maintained as semantic memories: plastic model $f(.; \theta_P)$ and the stable model $f(.; \theta_S)$. Finally, Reservoir sampling (Vitter, 1985) is employed to maintain a small episodic memory $\mathcal{M}$.

At each training step, the working model receives the training batch $X_b$ from the data stream and retrieves a random batch of exemplars $X_m$ from the episodic memory. This is then followed by the retrieval of optimal semantic information, i.e. the structural knowledge encoded in the semantic memories which account for the consolidation of feature space and adaptation of the decision boundaries of the previous tasks. The semantic memories are designed so that the plastic model has higher performance on recent tasks whereas the stable model prioritizes retaining information on the older tasks. Therefore, we would prefer to use the logits from the stable model $Z_S$ for older exemplars and the plastic model $Z_P$ for recent exemplars. As CLS-ER is a general incremental learning method, instead of using a hard threshold or task information, we opt for a simple task-agnostic approach of using the performance of the semantic memories on the exemplars as a selection criterion that empirically works well. For each exemplar, we select the replay logits $Z$ based on which model has the highest softmax score for the ground-truth class (lines 5-6 in Algorithm 1).

The selected replay logits from the semantic memories are then used to enforce a consistency loss on the working model so that it does not deviate from the already learned experiences. Hence, the working model is updated with a combination of the cross-entropy loss on the union of the data stream and episodic memory samples, $X$, and the consistency loss on the exemplars $X_m$,

$$\mathcal{L} = \mathcal{L}_{CE}(\sigma(f(X; \theta_W)), Y) + \lambda \mathcal{L}_{MSE}(f(X_m; \theta_W), Z) \tag{1}$$

---

**Algorithm 1** Complementary Learning System-Experience Replay Algorithm

---

**Input:** Data stream $\mathcal{D}$, Learning rate $\eta$, Consistency weight $\lambda$, Update rates $r_P$ and $r_S$, Decay parameters $\alpha_P$ and $\alpha_S$

**Initialize:** $\theta_W = \theta_P = \theta_S$

$\mathcal{M} \leftarrow \{\}$

1: **while** Training **do**
2:      $(X_b, Y_b) \sim \mathcal{D}$ and $(X_m, Y_m) \sim \mathcal{M}$
3:      $(X, Y) = \{(X_b, Y_b), (X_m, Y_m)\}$
4:      $Z_P, Z_S \leftarrow f(X_m; \theta_P), f(X_m; \theta_S)$              $\triangleright$ Select optimal semantic memory
5:      $Z \leftarrow Z_P$ **if** $\sigma(Z_P)^{(Y_m)} > \sigma(Z_S)^{(Y_m)}$ **else** $Z_S$
6:      $\mathcal{L} = \mathcal{L}_{CE}(\sigma(f(X; \theta_W)), Y) + \lambda \mathcal{L}_{MSE}(f(X_m; \theta_W), Z)$     $\triangleright$ Update working model
7:      $\theta_W \leftarrow \theta_W - \eta \nabla_{\theta_W} \mathcal{L}$
8:      $a, b \sim \mathcal{U}(0, 1)$                          $\triangleright$ Update semantic memories
9:      $\theta_P \leftarrow \alpha_p \theta_P + (1 - \alpha_P)\theta_W$ **if** $a < r_P$ **else** $\theta_P$
10:      $\theta_S \leftarrow \alpha_S \theta_S + (1 - \alpha_S)\theta_W$ **if** $b < r_S$ **else** $\theta_S$
11:      $\mathcal{M} \leftarrow Reservoir(\mathcal{M}, (X_b, Y_b))$          $\triangleright$ Update episodic memory (Algorithm 2)
     **return** $\theta_W, \theta_P, \theta_S$

---

where $\sigma$ is the softmax function, $\lambda$ the regularization parameter, and $\mathcal{L}_{MSE}$ the mean squared error loss used as consistency term.

After updating the working model, we stochastically update the plastic and stable models with rates $r_P$ and $r_S$ (note that $r_P > r_S$ so that the plastic model is updated more frequently). A stochastic rather than a deterministic approach is more biologically plausible (Maass, 2014; Arani et al., 2021) which reduces the overlap in the snapshots of the working model and leads to more diversity in semantic memories. The semantic memories are updated by taking an exponential moving average of the working model's weights (Tarvainen & Valpola, 2017) with decay parameters $\alpha_P$ and $\alpha_S$,

$$\theta_i = \alpha_i \theta_i + (1 - \alpha_i)\theta_W, \quad i \in \{P, S\} \tag{2}$$

Note that $\alpha_P \leq \alpha_S$ so that the plastic model mimics the rapid adaptation of information while the stable model mimics slow acquisition of structured knowledge. See Algorithm 1 for more details.

For inference, we use the stable model as it retains long-term memory across the tasks, consolidates structural knowledge, and learns efficient representations for generalization (Figure 1).

## 4    EXPERIMENTAL SETUP

To ensure a fair comparison of different CL methods under uniform experimental settings, we extended the Mammoth framework (Buzzega et al., 2020a) and unless stated otherwise, we follow the same training scheme (learning rate, batch sizes of incoming data and memory buffer, and the number of training epochs) as them for each of the evaluation settings. To find the optimal hyperparameters for CLS-ER, we run a grid search over $\lambda$, $\alpha_S$, $\alpha_P$, $r_S$, and $r_P$ on a small validation set. Sections C.4 and E show that our method is not highly sensitive to the particular choice of hyperparameters and different settings can attain similar performance. Also, because of the complementary nature of the components, we can often fix a set of parameters (e.g. $\lambda$, $\alpha_S$, $\alpha_P$ and $r_S$) and only finetune the remaining parameters (e.g. $r_P$) which facilitates hyperparameter tuning significantly.

Following Buzzega et al. (2020a), we employ a fully connected network with two hidden layers, each with 100 ReLU units on all the variants of the MNIST dataset and ResNet-18 (He et al., 2015) without pretraining for the other datasets. In all the settings, we use the SGD optimizer. We use random horizontal flip and random crop on both the stream and buffer samples for S-CIFAR-10, S-Tiny-ImageNet, and GCIL-CIFAR-100. The selected hyperparameters for each of the settings are provided in Table S4. Note that for the vast majority of datasets, we use uniform settings (lr, epochs, batch size, memory batch size, and $lambda$) across different buffer sizes and only slight modifications in the other hyperparameters which shows that our method does not require extensive finetuning for different memory budgets. For each of our experiments, we fix the order of the classes and report the average and one standard deviation of the mean test accuracy of all the tasks across 10 runs with different initializations. Section E provides further training and implementation details.

| Buffer | Method | Class-IL | | | Domain-IL | |
|---|---|---|---|---|---|---|
| | | S-MNIST | S-CIFAR-10 | S-Tiny-ImageNet | R-MNIST | P-MNIST |
| – | JOINT | $95.57_{\pm0.24}$ | $92.20_{\pm0.15}$ | $59.99_{\pm0.19}$ | $95.76_{\pm0.04}$ | $94.33_{\pm0.17}$ |
| | SGD | $19.60_{\pm0.04}$ | $19.62_{\pm0.05}$ | $7.92_{\pm0.26}$ | $67.66_{\pm8.53}$ | $40.70_{\pm2.33}$ |
| 200 | ER | $80.43_{\pm1.89}$ | $44.79_{\pm1.86}$ | $8.49_{\pm0.16}$ | $85.01_{\pm1.90}$ | $72.37_{\pm0.87}$ |
| | GEM | $80.11_{\pm1.54}$ | $25.54_{\pm0.76}$ | - | $80.80_{\pm1.15}$ | $66.93_{\pm1.25}$ |
| | iCaRL | $70.51_{\pm0.53}$ | $49.02_{\pm3.20}$ | $7.53_{\pm0.79}$ | - | - |
| | FDR | $79.43_{\pm3.26}$ | $30.91_{\pm2.74}$ | $8.70_{\pm0.19}$ | $85.22_{\pm3.35}$ | $74.77_{\pm0.83}$ |
| | GSS | $38.92_{\pm2.49}$ | $39.07_{\pm5.59}$ | - | $79.50_{\pm0.41}$ | $63.72_{\pm0.70}$ |
| | DER++ | $85.61_{\pm1.40}$ | $64.88_{\pm1.17}$ | $10.96_{\pm1.17}$ | $90.43_{\pm1.87}$ | $83.58_{\pm0.59}$ |
| | CLS-ER | $\mathbf{89.54}_{\pm0.21}$ | $\mathbf{66.19}_{\pm0.75}$ | $\mathbf{23.47}_{\pm0.80}$ | $\mathbf{92.26}_{\pm0.18}$ | $\mathbf{84.63}_{\pm0.40}$ |
| 500 | ER | $86.12_{\pm1.89}$ | $57.74_{\pm0.27}$ | $9.99_{\pm0.29}$ | $88.91_{\pm1.44}$ | $80.60_{\pm0.86}$ |
| | GEM | $85.99_{\pm1.35}$ | $26.20_{\pm1.26}$ | - | $81.15_{\pm1.98}$ | $76.88_{\pm0.52}$ |
| | iCaRL | $70.10_{\pm1.08}$ | $47.55_{\pm3.95}$ | $9.38_{\pm1.53}$ | - | - |
| | FDR | $85.87_{\pm4.04}$ | $28.71_{\pm3.23}$ | $10.54_{\pm0.21}$ | $89.67_{\pm1.63}$ | $83.18_{\pm0.53}$ |
| | GSS | $49.76_{\pm4.73}$ | $49.73_{\pm4.78}$ | - | $81.58_{\pm0.58}$ | $76.00_{\pm0.87}$ |
| | DER++ | $91.00_{\pm1.49}$ | $72.70_{\pm1.36}$ | $19.38_{\pm1.41}$ | $92.77_{\pm1.05}$ | $88.21_{\pm0.39}$ |
| | CLS-ER | $\mathbf{92.05}_{\pm0.32}$ | $\mathbf{75.22}_{\pm0.71}$ | $\mathbf{31.03}_{\pm0.56}$ | $\mathbf{94.06}_{\pm0.07}$ | $\mathbf{88.30}_{\pm0.14}$ |
| 5120 | ER | $93.40_{\pm1.29}$ | $82.47_{\pm0.52}$ | $27.40_{\pm0.31}$ | $93.45_{\pm0.56}$ | $89.90_{\pm0.13}$ |
| | GEM | $95.11_{\pm0.87}$ | $25.26_{\pm3.46}$ | - | $88.57_{\pm0.40}$ | $87.42_{\pm0.95}$ |
| | iCaRL | $70.60_{\pm1.03}$ | $55.07_{\pm1.55}$ | $14.08_{\pm1.92}$ | - | - |
| | FDR | $87.47_{\pm3.15}$ | $19.70_{\pm0.07}$ | $28.97_{\pm0.41}$ | $94.19_{\pm0.44}$ | $90.87_{\pm0.16}$ |
| | GSS | $89.39_{\pm0.75}$ | $67.27_{\pm4.27}$ | - | $85.24_{\pm0.59}$ | $82.22_{\pm1.14}$ |
| | DER++ | $95.30_{\pm1.20}$ | $85.24_{\pm0.49}$ | $39.02_{\pm0.97}$ | $\mathbf{94.65}_{\pm0.33}$ | $\mathbf{92.26}_{\pm0.17}$ |
| | CLS-ER | $\mathbf{95.73}_{\pm0.11}$ | $\mathbf{86.78}_{\pm0.17}$ | $\mathbf{46.74}_{\pm0.31}$ | $94.25_{\pm0.06}$ | $92.03_{\pm0.05}$ |

Table 1: Comparison with prior works on Class-IL and Domain-IL settings. The baseline results are from Buzzega et al. (2020a) (- indicates the experiments that the authors were unable to run).

## 5 EMPIRICAL EVALUATION

There are a plethora of evaluation protocols in the CL literature, each of which biases the evaluation towards a certain approach (Farquhar & Gal, 2018; Mi et al., 2020; van de Ven & Tolias, 2019). It is therefore of utmost importance to conduct an extensive and robust evaluation over different CL settings to gauge the versatility of the method. Details of the datasets used in each CL setting are provided in Section A. We compare our method with the state-of-the-art rehearsal-based approaches on various CL settings and memory budgets under uniform experimental settings. *SGD* refers to standard training and *JOINT* provides an upper bound given by training all tasks jointly.

**Class Incremental Learning (Class-IL):** refers to the CL scenario where new classes are added with each subsequent task and the agent must learn to distinguish not only amongst the classes within the current task but also across previous tasks. Class-IL measures how well the method can learn general representations, accumulate, consolidate, and transfer the acquired knowledge to learn efficient representations and decision boundaries for all the classes seen so far.

Table 1 provides the comparison with six rehearsal-based approaches on Class-IL settings with varying datasets and task length complexities. CLS-ER provides the highest performance in all of these scenarios. In particular, as the dataset complexity and number of tasks increase from S-MNIST to S-Tiny-ImageNet, the performance gap between CLS-ER and DER++ increases considerably. Especially, with a smaller memory budget, CLS-ER is able to retain more information than other methods. In the most challenging setting, S-Tiny-ImageNet with 200 buffer size, CLS-ER provides a percentage gain of $176\%$ and $114\%$ over the baseline ER and the current state-of-the-art DER++, respectively. The results demonstrate the capability of CLS-ER to efficiently accumulate and retain knowledge over longer sequences under complex and memory restrictive scenarios.

We believe that the performance gains over DER++ highlight a key component of an efficient CL agent: the ability to consolidate previously acquired knowledge. DER++ fails to account for the consolidation of feature space and adaptation of the decision boundaries of the previous tasks. Therefore, constraining the model to match the sub-optimal logits might hamper the consolidation of knowledge. This becomes more prominent as the number of classes in each task, the sequence

| JOINT | SGD | Buffer | ER | MER | GSS | DER++ | CLS-ER |
|---|---|---|---|---|---|---|---|
| | | 200 | $49.27_{\pm2.25}$ | $48.58_{\pm1.07}$ | $43.92_{\pm2.43}$ | $54.16_{\pm3.02}$ | $\mathbf{66.37}_{\pm0.83}$ |
| $82.98_{\pm3.24}$ | $19.09_{\pm0.69}$ | 500 | $65.04_{\pm1.53}$ | $62.21_{\pm1.36}$ | $54.45_{\pm3.14}$ | $69.62_{\pm1.59}$ | $\mathbf{75.70}_{\pm0.41}$ |
| | | 1000 | $75.18_{\pm1.50}$ | $70.91_{\pm0.76}$ | $63.84_{\pm2.09}$ | $76.03_{\pm1.61}$ | $\mathbf{79.54}_{\pm0.34}$ |

Table 2: Comparison with prior works on MNIST-360 test set. The baseline results are from Buzzega et al. (2020a).

| Distribution | Uniform | | | Longtail | | |
|---|---|---|---|---|---|---|
| JOINT | $58.36_{\pm1.02}$ | | | $56.94_{\pm1.56}$ | | |
| SGD | $12.67_{\pm0.24}$ | | | $22.88_{\pm0.53}$ | | |
| Buffer | 200 | 500 | 1000 | 200 | 500 | 1000 |
| ER | $16.40_{\pm0.37}$ | $28.21_{\pm0.69}$ | $31.98_{\pm0.72}$ | $19.27_{\pm0.77}$ | $20.30_{\pm0.63}$ | $34.13_{\pm0.83}$ |
| DER++ | $18.84_{\pm0.60}$ | $32.92_{\pm0.74}$ | $38.95_{\pm0.56}$ | $26.94_{\pm1.27}$ | $25.82_{\pm0.83}$ | $33.64_{\pm0.88}$ |
| CLS-ER | $\mathbf{25.06}_{\pm0.81}$ | $\mathbf{36.34}_{\pm0.59}$ | $\mathbf{39.69}_{\pm0.66}$ | $\mathbf{28.54}_{\pm0.87}$ | $\mathbf{28.63}_{\pm0.68}$ | $\mathbf{39.52}_{\pm0.91}$ |

Table 3: Comparison with prior works on GCIL-CIFAR-100 dataset.

length, and the cross-task resemblance increase. For instance, for DER++, replaying a sample from Task-1 when training on S-Tiny-ImageNet Task-10, the reference logit values which are used to enforce the consistency are from a model representation state which has not considered how to distinguish the 20 classes in Task-1 from 80 additional classes which are visually and semantically similar. It stands to reason that the optimal representation space and subsequently the decision boundaries for the classes in Task-1 would drift considerably when required to distinguish between 80 additional classes as well. Therefore, the local information provided by the sub-optimal saved logits in DER++ fails to provide the global context required for consolidating knowledge. CLS-ER, on the other hand, extracts logits from the semantic memories which consolidate knowledge across the tasks, and hence the working model receives more optimal feedback.

**Domain Incremental Learning (Domain-IL):** refers to the CL scenario where the classes remain the same in subsequent tasks but the input distribution changes. We consider R-MNIST where each task contains digits rotated by a fixed angle and P-MNIST which applies a fixed random permutation to the pixels for each task. Table 1 shows that CLS-ER provides generalization gains under both settings, particularly for lower memory budget, and performs on par with DER++ on 5120 buffer size. We attribute this to the consolidated soft targets from the semantic memories which provide relational information about the classes from a global context compared to the local information in DER++. This enables our method to maintain the similarity structure across sequences effectively.

**General Incremental Learning (GIL):** Class-IL and Domain-IL fail to assimilate the challenges in the real-world setting where the task boundaries are blurry, and classes can reappear and have different distributions. The CL method has to consider the sample efficiency, challenge of imbalanced data, and efficient knowledge transfer in addition to preventing catastrophic forgetting. We consider two GIL settings: MNIST-360 (Buzzega et al., 2020a) exposes the model to both sharp (changes in class) and smooth (rotation of digits) distribution shifts. This requires the CL method to tackle the challenges of class-IL as well as domain-IL. The Generalized Class Incremental Learning (GCIL; Mi et al. (2020)) is the closest to the real-world scenario as it utilizes probabilistic modeling to sample the classes and data distributions in each task. The number of classes in each task is not fixed, the classes can overlap and the sample size for each class can vary.

Table 2 shows that CLS-ER provides considerable performance gains on the challenging MNIST-360, particularly with a low memory budget. Similarly, Table 3 demonstrates the effectiveness of CLS-ER on GCIL-CIFAR-100 under both uniform and imbalanced class samples. Both of these settings involve recurring classes in subsequent sequences which makes the transfer of knowledge from previous occurrences important. The performance gap between CLS-ER and DER++ in the recurring classes setting alludes to another shortcoming of saving logits from the previous state. Consider the case where class c appears in sequence (Seq)-1 with 20 samples, and then subsequently in Seq-5 with 200 samples. In the following sequences, DER++ uses exemplars from class c saved in Seq-1 with sub-optimal logits from the model state which was attained with only 20 samples and fails to take advantage of the better learned representations with additional data in Seq-5. CLS-ER,

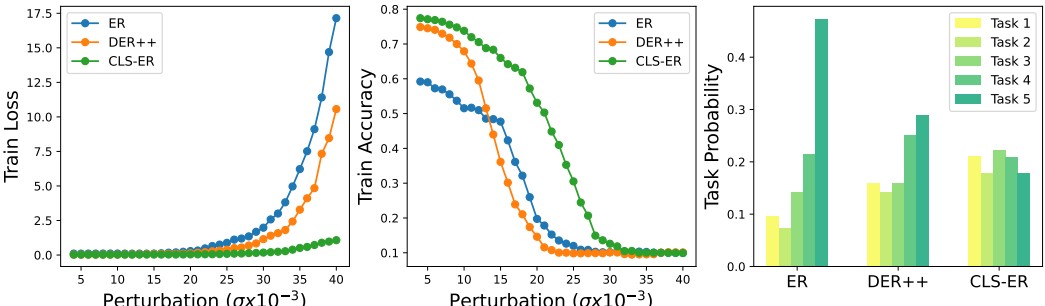

Figure 3: Model characteristics analyses of different methods trained on S-CIFAR-10 with 500 buffer size. The Left and middle figures show the training loss and accuracy under varying Gaussian noise added to the weights of each layer of the model. CLS-ER is considerably less sensitive to perturbations, suggesting convergence to flatter minima. The right figure shows the task probabilities. CLS-ER effectively mitigates the bias to the recent tasks and provides a more uniform probability of being predicted for the classes over the tasks even very early ones.

on the other hand, is able to take advantage of the additional samples and provide feedback from the improved learned representations. Moreover, the considerable performance improvement in the longtail setting shows that CLS-ER is more robust to class imbalance

Note that the MNIST-based settings can be considered under the online CL setting (see Section A.4) as we only pass through the data once for each task and the performance of CLS-ER on these settings demonstrates its potential as an efficient method for online CL.

## 6 Model Characteristics

We analyze CLS-ER and provide some insights into the characteristics of the proposed approach which enables it to learn effectively under challenging CL scenarios. In the subsequent analyses, we compare CLS-ER with the baseline ER and DER++.

### 6.1 Convergence to Flatter Minima

Due to the non-convexity of the loss landscape, there can be multiple solutions to the optimization objective, however, the local geometry at the convergence point can affect the generalization of the model. Solutions that reside in wide valleys instead of narrow crevices generalize better (Chaudhari et al., 2019; Hochreiter & Schmidhuber, 1997; Keskar et al., 2016) as the predictions do not change drastically with small perturbations. A CL model which converges to flatter minima has more flexibility to explore the neighboring parameter space to optimize on the new task without drastically increasing the loss on the previous tasks. Following the analysis in Zhang et al. (2018), we add independent Gaussian noise of increasing strength to the parameters of the trained model and analyze the change in accuracy and loss across the training samples. Figure 3 shows that CLS-ER is significantly less sensitive to perturbations compared to ER and DER++. CLS-ER also retains performance for a longer period and its performance drops more smoothly. These results suggest that the fast and slow adaptation of information in CLS-ER can guide the optimization to wider valleys.

### 6.2 Task Probabilities

Because of the sequential nature of CL, an implicit bias is induced towards the current task (Wu et al., 2019). A number of CL methods employ explicit techniques to reduce this bias (Hou et al., 2019; Wu et al., 2019), however, they utilize the task boundaries which is counterproductive for general incremental learning. We believe that the efficient knowledge consolidation in CLS-ER through the semantic memories can implicitly mitigate the bias towards recent tasks. We follow the analysis performed in Buzzega et al. (2020b) to observe the probability of each task being predicted at the end of the training. For each sample in the test dataset, we take the softmax output and then

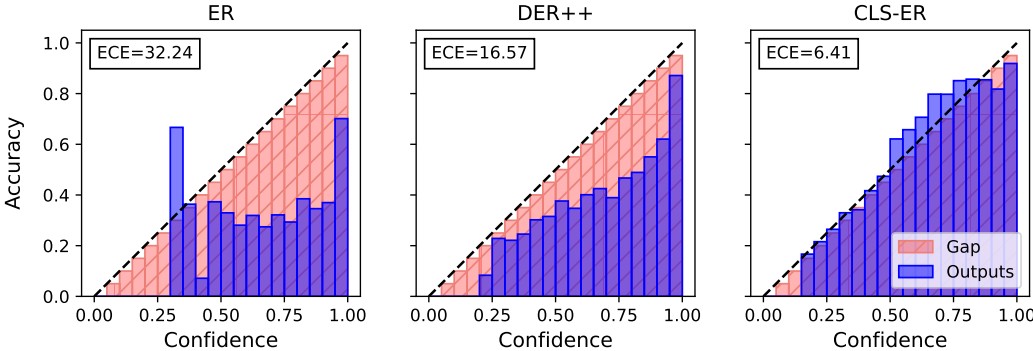

Figure 4: Reliability plots for different methods on S-CIFAR-10 with 500 buffer size. CLS-ER results in considerably better-calibrated models and hence more reliable predictions. For other buffer sizes and S-TinyImageNet see Figures S5 and S6.

average the probabilities of the associated classes for each task across the dataset. We normalize the values and report the probability of each task being predicted. Figure 3 (right plot) shows that CLS-ER is able to maintain a more uniform prediction probability across all the tasks over a long sequence. Figures S3 and S4 shows similar results for other buffer sizes and S-TinyImageNet.

## 6.3 MODEL CALIBRATION

Model calibration refers to the accuracy with which the scores provided by the model reflect its predictive uncertainty. The class probabilities predicted by DNNs are uncalibrated, often tending towards over-confidence which is detrimental to the reliability of the model's prediction (Guo et al., 2017). This is even more pronounced in CL where the models tend to be biassed towards recent tasks. Following Guo et al. (2017), we provide the reliability diagrams (model accuracy as a function of its prediction confidence) and the Expected Calibration Error (ECE; a weighted average over the absolute difference between accuracy and confidence). Figure 4 shows the remarkable ability of CLS-ER to provide well-calibrated models without the application of any calibration technique.

Note that these characteristics are complementary in nature: convergence to flatter minima allows our method to remain in the vicinity of optimal parameters for previous tasks when adapting to the new task, this leads to more uniform performance across tasks which can improve the task probabilities, and since the model is not too biased towards the current task, the model can provide reliable prediction across the tasks which improve the calibration. Additional characteristics analyses on different datasets and buffer sizes are provided in Appendix. We observe that our model's behavior is consistent across varying datasets and buffer sizes.

## 7 CONCLUSION

We proposed a novel dual memory experience replay method based on the complementary learning systems theory in the brain. Our method maintains long-term and short-term semantic memories which are utilized to effectively replay the neural activities of the episodic memories and align the decision boundary of the working model for efficient knowledge consolidation. We demonstrated the effectiveness of our approach on benchmark datasets as well as more challenging general incremental learning scenarios and achieved the new state-of-the-art in the vast majority of the continual learning settings. We further showed that CLS-ER converges to flatter minima, mitigates the bias towards recent tasks, and provides a well-calibrated high-performance model. Our strong empirical results motivate further study into mimicking the complementary learning system in the brain more faithfully to enable optimal continual learning in DNNs.

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

# A  CONTINUAL LEARNING SETTINGS

There are a plethora of evaluation protocols in the CL literature, each of which biases the evaluation towards a certain approach (Farquhar & Gal, 2018; Mi et al., 2020; Shim et al., 2020; van de Ven & Tolias, 2019). It is therefore of utmost importance to conduct an extensive and robust evaluation to gauge the versatility of the method. We believe that adhering to the key desiderata as suggested in Farquhar & Gal (2018) would help the CL community immensely in moving towards a robust evaluation of methods. An experimental protocol that trains the method on a long sequence of tasks where the boundaries between the tasks are not distinct and the tasks themselves are not disjoint and the method does not make sure of task boundaries during training or testing can be considered as adhering to all five desiderata. Our work focuses on the aforementioned setting which can be considered as General Incremental Learning (GIL) setting. Here, we provide a broad categorization of these evaluation protocols which test different aspects of CL.

## A.1  CLASS INCREMENTAL LEARNING (CLASS-IL)

Class-IL refers to the CL scenario where new classes are added with each subsequent task and the agent must learn to distinguish not only amongst the classes within the current task but also across previous tasks. Class-IL measures how well the method can learn general representations, accumulate, consolidate, and transfer the acquired knowledge to learn efficient representations and decision boundaries for all the classes seen so far. Following Buzzega et al. (2020a); De Lange et al. (2019); Zenke et al. (2017), we consider the common benchmark datasets MNIST (LeCun et al., 1998) (**S-MNIST**), CIFAR-10 (Krizhevsky et al., 2009) (**S-CIFAR-10**) and Tiny-ImageNet (Pouransari & Ghili, 2015) (**S-Tiny-ImageNet**) which are split into 5, 5, and 10 tasks each including 2, 2, and 20 classes respectively. These represent Class-IL settings of increasing dataset complexity as well as longer sequences. While it is an important and challenging benchmark, it assumes that each subsequent task will have the same number of disjoint classes and have uniform samples for each class which is not representative of real-world scenarios. We do not consider the related Task Increment Learning (Task-IL) setting as it assumes the availability of task labels at both training and inference which cannot truly be considered as a CL task (Farquhar & Gal, 2018).

## A.2  DOMAIN INCREMENTAL LEARNING (DOMAIN-IL)

Domain-IL refers to the CL scenario where the classes remain the same in each subsequent task but the input distribution changes. We consider **Rotated-MNIST** (Lopez-Paz & Ranzato, 2017) (R-MNIST) where each task contains digits rotated by a fixed angle between 0 and 180 degrees and **Permuted-MNIST** (Kirkpatrick et al., 2016) (P-MNIST) which applies a fixed random permutation to the pixels for each task. Though we provide the results for Permuted MNIST for completion, we share the opinion by Farquhar & Gal (2018) that it should not be considered as a benchmark dataset as it violates the cross-task resemblance desiderata and deviates from the goal of continual learning.

## A.3  GENERAL INCREMENTAL LEARNING (GIL)

The aforementioned CL scenarios fail to assimilate the challenges in the real world, setting where the task boundaries are blurry and the learning agent must rather learn from a continuous stream of data where classes can reappear and have different data distributions. The CL method must deal with the issues of sample efficiency, imbalanced classes, and efficient transfer of knowledge in addition to preventing catastrophic forgetting. To test the efficacy of our method in this challenging setting, we consider two GIL evaluation protocols. **MNIST-360** (Buzzega et al., 2020a) models a stream of data which presents batches of two consecutive MNIST images with each sample rotated at an increasing angle and the sequence is repeated three times. This exposes the model to both a sharp distribution shift when the class changes and a smooth rotational distribution shift. However, the number of classes in each task and the samples are uniform. The Generalized Class Incremental Learning (GCIL) (Mi et al., 2020) utilizes probabilistic modeling to sample the classes and data distributions in each task. Hence, the number of classes in each task is not fixed, the classes can overlap and the sample size for each class can vary. Following Mi et al. (2020), we use GCIL on CIFAR-100 (Krizhevsky et al., 2009) dataset (**GCIL-CIFAR-100**), set the number of samples

and maximum number of classes per task to 1000 and 50 respectively, number of tasks to 20, and evaluate on both uniform and longtail (imbalanced) sample distribution.

### A.4 ONLINE CONTINUAL LEARNING

Online continual learning refers to the challenging scenario where a stream of samples is only seen once and is non-iid (Mai et al., 2022; Aljundi et al., 2019). The common approach in the literature is to use the single-epoch protocol where the network is trained on each task in the sequence for only one epoch and there are no additional passages over data. As we aim to position CLS-ER as a general incremental learning method, we are also interested in the online continual learning setting. However, similar to Buzzega et al. (2020a), we also believe that the dataset complexity needs to be considered when setting the number of epochs to disentangle the effect of catastrophic forgetting from underfitting and share their suggestion that future CL works should strive for realism by designing experimental settings which are in line with the guidelines of General Continual Learning (Farquhar & Gal, 2018) which is the goal of our study rather than adopting the single-epoch protocol. For the MNIST-based settings, we use only one epoch per task as it is sufficient for the SGD baseline to learn the single task well. And for the more complex settings, we increase the number of epochs: 50 epochs for Sequential CIFAR-10 and Sequential Tiny-ImageNet and 100 epochs for GCIL-CIFAR-100.

We would also like to emphasize that the experiments on MNIST based settings (S-MNIST, R-MNIST, P-MNIST, and MNIST-360) can be considered as online continual learning settings as we only train the network for 1 epoch, and thereby the model only sees the data for each task once. CLS-ER's performance in these settings demonstrates its potential for the challenging online continual learning setting.

## B RESERVOIR SAMPLING

Here, we provide the algorithm for the *Reservoir Sampling* for maintaining a fixed-size memory buffer. Reservoir sampling takes in a data stream of unknown length and assigns equal probability to each sample for being represented in the memory buffer ($\mathcal{M}$) with a fixed budget size ($\mathcal{B}$). Sampling and replacement are done at random and no priority is assigned to the samples being added or replaced from the memory buffer.

---

**Algorithm 2** Reservoir Sampling Algorithm

---

    **Input:** Memory Buffer $\mathcal{M}$, Memory Budget $\mathcal{B}$, Number of seen examples $N$, Selected example $(x, y)$

1: **if** $\mathcal{B} > N$ **then**                                          ▷ Memory is not full
2:     $\mathcal{M}[N] \leftarrow (x, y)$
3: **else**                                               ▷ Select a sample to remove
4:     $\nu = randomInteger(min = 0, max = N)$
5:     **if** $\nu < \mathcal{B}$ **then**
6:         $\mathcal{M}[\nu] \leftarrow (x, y)$
    **return** $\mathcal{M}$

---

## C ADDITIONAL RESULTS

In this section, we provide additional experimental results and analysis of the behavior of the model.

### C.1 CLS-ER COMPONENTS PERFORMANCE

CLS-ER involves the interplay between the working model and the two semantic memories: the plastic and stable models. While we use the stable model for final inference, here we provide the performance of each of these individual components to provide further insights into the workings of our method. Table S1 shows the corresponding performance of the working model and plastic model for each of our experimental settings. We can see that the stable model can effectively consolidate

| Dataset | Buffer | Stable Model | Working Model | Plastic Model |
|---|---|---|---|---|
| S-MNIST | 200 | $89.54_{\pm0.21}$ | $89.32_{\pm0.23}$ | $89.52_{\pm0.21}$ |
| | 500 | $92.05_{\pm0.30}$ | $91.61_{\pm0.47}$ | $92.04_{\pm0.33}$ |
| | 5120 | $95.73_{\pm0.10}$ | $95.65_{\pm0.15}$ | $95.73_{\pm0.12}$ |
| S-CIFAR-10 | 200 | $66.19_{\pm0.75}$ | $50.09_{\pm1.48}$ | $62.68_{\pm1.94}$ |
| | 500 | $75.22_{\pm0.71}$ | $63.09_{\pm1.12}$ | $71.32_{\pm0.89}$ |
| | 5120 | $86.78_{\pm0.17}$ | $85.00_{\pm0.33}$ | $86.77_{\pm0.17}$ |
| S-Tiny-ImageNet | 200 | $23.47_{\pm0.80}$ | $9.97_{\pm0.18}$ | $17.19_{\pm0.71}$ |
| | 500 | $31.03_{\pm0.56}$ | $15.35_{\pm0.34}$ | $27.16_{\pm0.43}$ |
| | 5120 | $46.74_{\pm0.31}$ | $41.39_{\pm0.39}$ | $47.10_{\pm0.42}$ |
| R-MNIST | 200 | $92.26_{\pm0.18}$ | $89.37_{\pm0.47}$ | $89.99_{\pm0.43}$ |
| | 500 | $94.06_{\pm0.07}$ | $93.24_{\pm0.14}$ | $93.52_{\pm0.09}$ |
| | 5120 | $94.25_{\pm0.06}$ | $94.28_{\pm0.08}$ | $94.37_{\pm0.06}$ |
| P-MNIST | 200 | $84.63_{\pm0.40}$ | $84.33_{\pm0.45}$ | $84.54_{\pm0.41}$ |
| | 500 | $88.30_{\pm0.14}$ | $88.12_{\pm0.16}$ | $88.25_{\pm0.14}$ |
| | 5120 | $92.03_{\pm0.05}$ | $91.96_{\pm0.06}$ | $92.02_{\pm0.05}$ |
| MNIST-360 | 200 | $66.37_{\pm0.83}$ | $55.59_{\pm1.74}$ | $60.60_{\pm1.41}$ |
| | 500 | $75.70_{\pm0.41}$ | $72.70_{\pm0.80}$ | $75.03_{\pm0.37}$ |
| | 1000 | $79.54_{\pm0.34}$ | $78.39_{\pm0.69}$ | $79.16_{\pm0.42}$ |
| GCIL-CIFAR-100 (Uniform) | 200 | $33.15_{\pm2.80}$ | $31.74_{\pm2.72}$ | $32.70_{\pm2.78}$ |
| | 500 | $37.01_{\pm1.67}$ | $35.89_{\pm1.69}$ | $36.18_{\pm1.68}$ |
| | 1000 | $41.09_{\pm1.58}$ | $40.44_{\pm1.80}$ | $40.70_{\pm1.66}$ |
| GCIL-CIFAR-100 (Longtail) | 200 | $29.57_{\pm3.80}$ | $28.19_{\pm3.90}$ | $29.12_{\pm3.89}$ |
| | 500 | $33.26_{\pm3.66}$ | $32.22_{\pm3.79}$ | $32.95_{\pm3.70}$ |
| | 1000 | $39.21_{\pm3.46}$ | $38.51_{\pm3.55}$ | $38.84_{\pm3.52}$ |

Table S1: CLS-ER components performance analysis for each of the experimental setting.

knowledge across the tasks and therefore provide the highest mean performance for the vast majority of the settings. Figures S1 and S2 further shows how the task-wise performance (on test set) of each of the component varies as subsequent tasks are learned. The stable model retains the performance on previous tasks while the plastic model adapts better to the recent task. Both these models provide feedback to the working model which in turn improves the plastic and stable model.

## C.2 TASK PROBABILITIES

To test the effectiveness of our method in mitigating the bias towards recent tasks, we provide the task probabilities of the models trained with different buffer sizes on S-CIFAR-10 and S-Tiny-ImageNet. Figures S3 and S4 show that CLS-ER consistently achieves more uniform task probabilities compared to ER and DER++ and effectively mitigates the bias towards the last task.

## C.3 MODEL CALIBRATION

To further test the consistency of CLS-ER in providing well-calibrated models and the impact of the buffer size, we evaluate the calibration of models trained with different buffer sizes on S-CIFAR-10 and S-Tiny-ImageNet. Figures S5 and S6 show that CLS-ER consistently provides better calibrated models compared to ER and DER++. Remarkably, for both the datasets, on lower buffer sizes, the difference in Expected Calibration Error (ECE) is considerable. This demonstrates the capability of CLS-ER to train high-performance and reliable models under challenging conditions.

## C.4 EFFECT OF HYPERPARAMETERS

The interaction between the three components of CLS-ER is complementary. Table S3 shows how the performance of each component is affected under different hyperparameter settings. We can draw the following conclusions from the results. The performance improvement in the plastic and stable model is reflected in the working model and the best performance is seen in cases where both the semantic memories are performing well (albeit the focus on tasks is different). This highlights

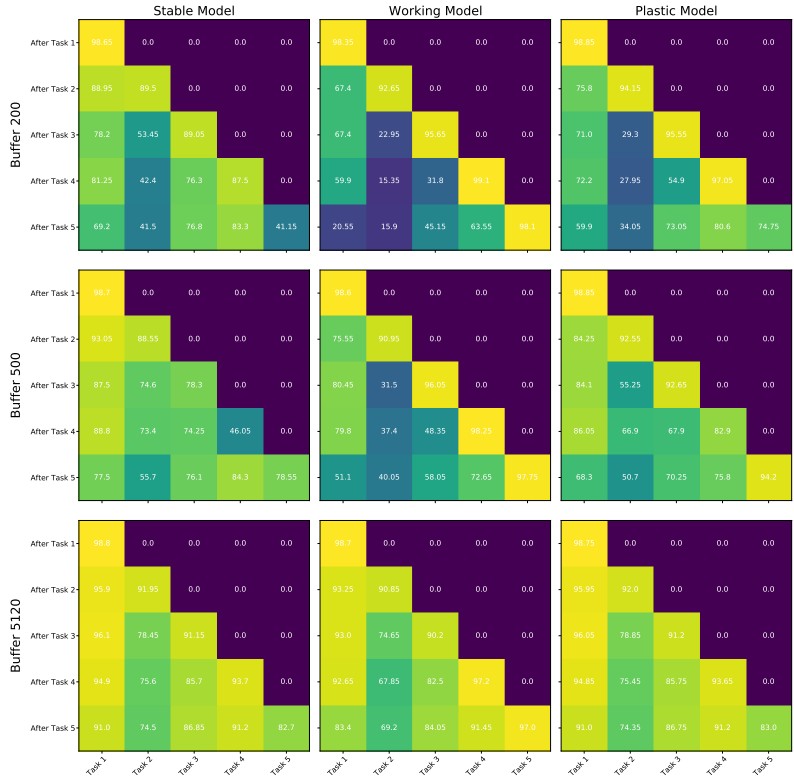

Figure S1: Test set task-wise performance for the individual models on S-CIFAR-10 with different buffer sizes. The task-wise performance (x-axis) is evaluated at the end of training of each task (y-axis) to evaluate how it is affected as training progresses.

the crucial role of both memories in enabling CLS-ER to learn efficiently. For a fixed $r_S$ value, the final performance of the stable model is affected considerably by the performance of the plastic model. The method is not highly sensitive to the particular choice of hyperparameters as different settings can attain similar performance. Because of the complementary nature of the components, we can often fix a set of parameters (e.g. $\lambda$, $\alpha_S$, $\alpha_S$ and $r_S$) and only finetune the remaining parameters (e.g. $r'_P$) which facilitates hyperparameter tuning significantly.

## D COMPARISON WITH A SINGLE SEMANTIC MEMORY

CLS-ER employs two semantic memories as we aim to mimic the fast and slow learning mechanisms in the hippocampus and neocortex respectively. Here we compare our method with a single semantic memory (Mean-ER) and Table S2 shows that while it still performs admirably compared to the other CL methods, the dual semantic memories in CLS-ER provides additional performance gains especially on the complex datasets under the challenging lower memory buffer settings and has a much lower variance. We attribute this to the failure of Mean-ER in maintaining the performance on both the recent and earlier tasks together i.e there is an inherent trade-off as tuning the semantic memory to adapt to the recent changes comes at the cost of performance on earlier tasks and vice versa. CLS-ER efficiently tackles this trade-off by maintaining two specialized long-term and short-term memories. The performance of Mean-ER, however, provides further evidence for the benefits of using consolidated information for memory replay.

Note that for a fair comparison, we use the same hyperparameter search space as CLS-ER for finding the optimal parameters for Mean-ER and report the average and 1 std of 10 runs with different initializations using the best parameters for each setting. Table S6 provides the chosen hyperparameters. For inference, similar to CLS-ER, we use the EMA-weighted model (semantic memory) for Mean-ER.

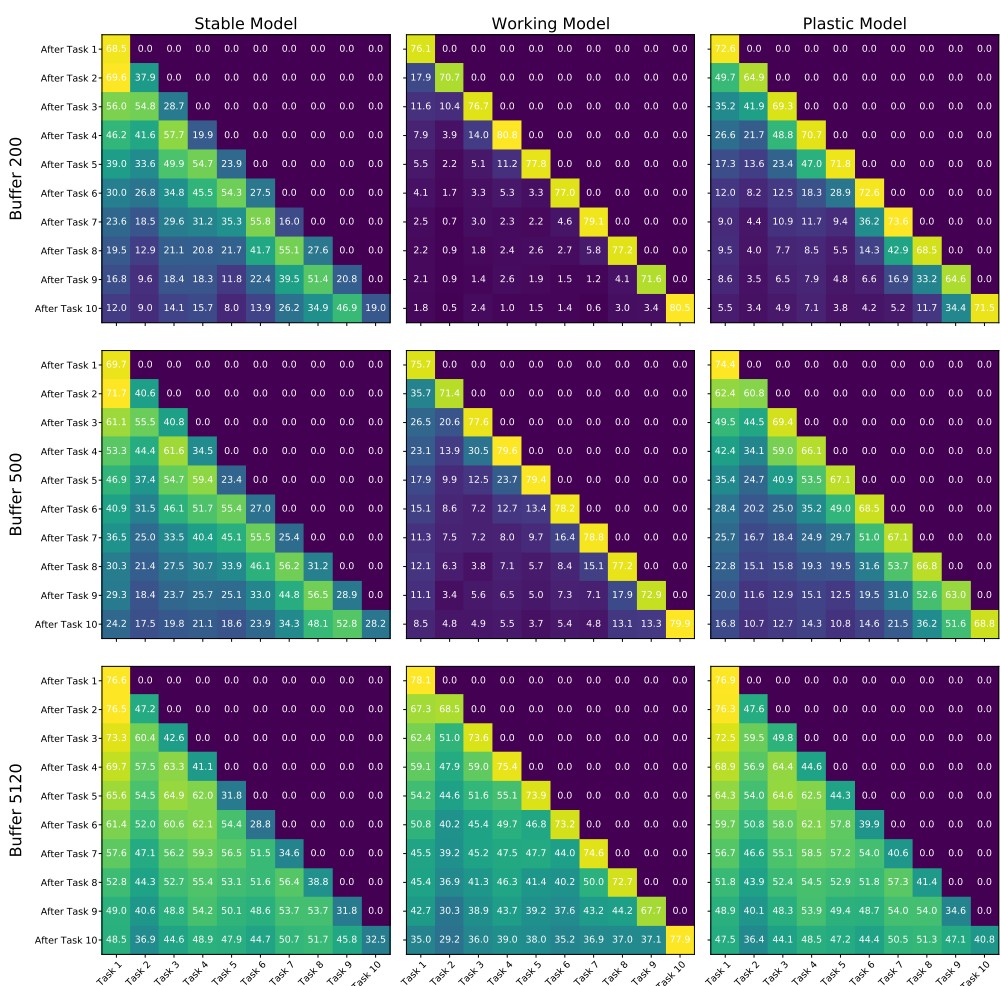

Figure S2: Test set task-wise performance for the individual models on S-Tiny-ImageNet with different buffer sizes. The task-wise performance (x-axis) is evaluated at the end of training of each task (y-axis) to evaluate how it is affected as training progresses.

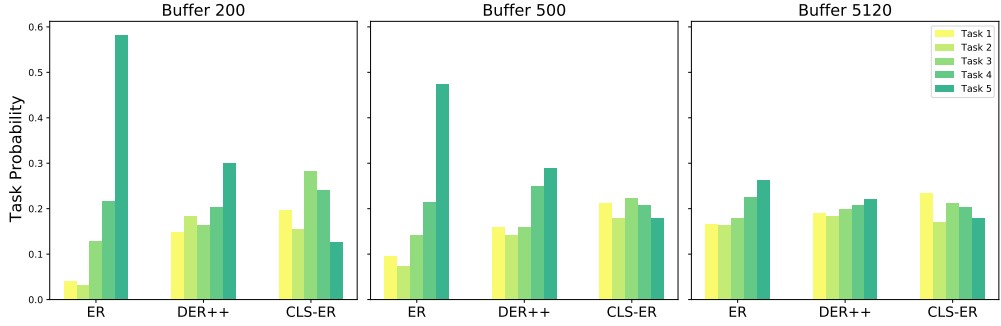

Figure S3: Task probabilities for different methods on S-CIFAR-10 with varying memory budget.

# E  TRAINING AND IMPLEMENTATION DETAILS

For a fair comparison, we aim to keep the experimental settings close to the current state-of-the-art DER++ (Buzzega et al., 2020a) as much as possible to disassociate the effect of the training

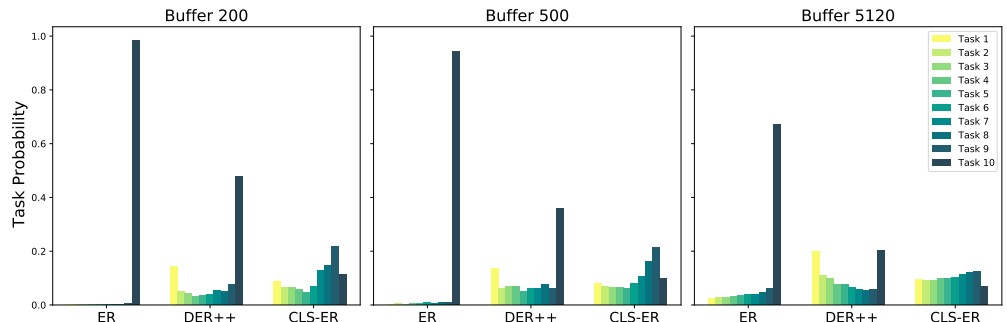

Figure S4: Task probabilities for different methods on S-Tiny-ImageNet with varying memory budget.

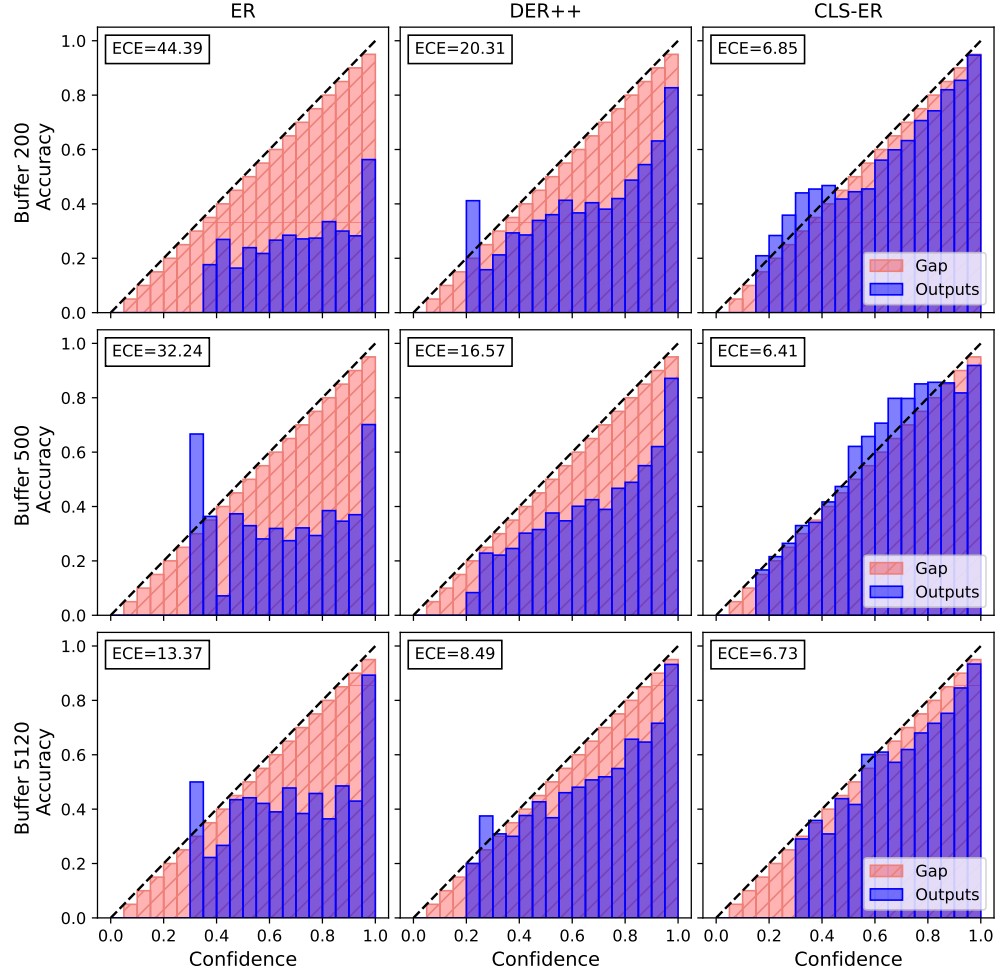

Figure S5: Reliability plots for the different methods on S-CIFAR-10 with varying memory budget.

schedule. We use the same optimizer, the number of epochs, batch size, and memory batch size as DER++. For S-Tiny-ImageNet, we reduce the number of epochs to 50 from 100 used by DER++ as our method can learn efficiently with fewer epochs, and quickly acquiring new knowledge is preferred for CL. Similar to DER++, we finetune the memory batch size for S-MNIST and MNIST-360. We select the hyperparameters for each of the experimental setting using a small validation set,

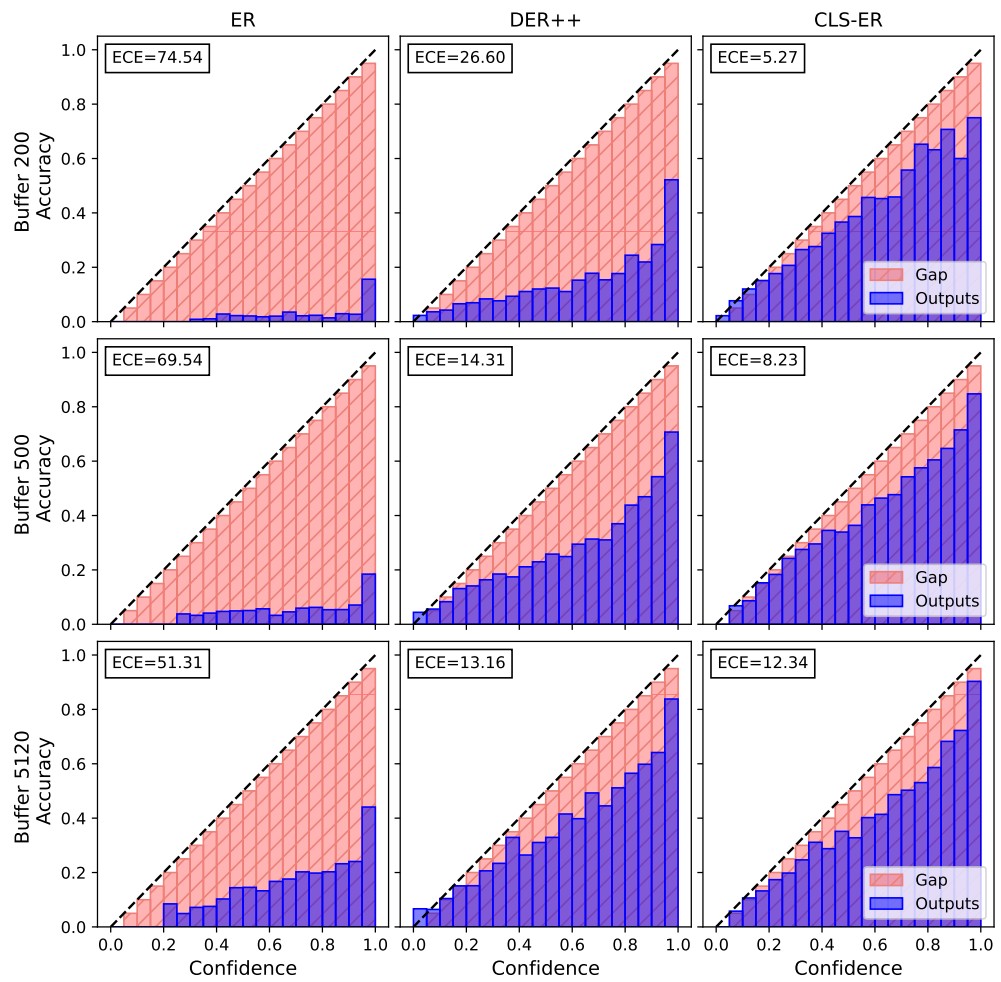

Figure S6: Reliability plots for the different methods on S-Tiny-ImageNet with varying memory budget.

| Buffer | Method | Class-IL | | | Domain-IL | |
|---|---|---|---|---|---|---|
| | | **S-MNIST** | **S-CIFAR-10** | **S-Tiny-ImageNet** | **R-MNIST** | **P-MNIST** |
| – | JOINT | $95.57_{\pm0.24}$ | $92.20_{\pm0.15}$ | $59.99_{\pm0.19}$ | $95.76_{\pm0.04}$ | $94.33_{\pm0.17}$ |
| | SGD | $19.60_{\pm0.04}$ | $19.62_{\pm0.05}$ | $7.92_{\pm0.26}$ | $67.66_{\pm8.53}$ | $40.70_{\pm2.33}$ |
| 200 | Mean-ER | $88.32_{\pm0.65}$ | $61.88_{\pm2.43}$ | $17.68_{\pm1.65}$ | $92.10_{\pm1.07}$ | $83.28_{\pm0.68}$ |
| | CLS-ER | $\mathbf{89.54}_{\pm0.21}$ | $\mathbf{66.19}_{\pm0.75}$ | $\mathbf{23.47}_{\pm0.80}$ | $\mathbf{92.26}_{\pm0.18}$ | $\mathbf{84.63}_{\pm0.40}$ |
| 500 | Mean-ER | $91.79_{\pm0.23}$ | $70.40_{\pm1.21}$ | $24.97_{\pm0.80}$ | $92.78_{\pm0.44}$ | $87.73_{\pm0.39}$ |
| | CLS-ER | $\mathbf{92.05}_{\pm0.32}$ | $\mathbf{75.22}_{\pm0.71}$ | $\mathbf{31.03}_{\pm0.56}$ | $\mathbf{94.06}_{\pm0.07}$ | $\mathbf{88.30}_{\pm0.14}$ |
| 5120 | Mean-ER | $95.57_{\pm0.18}$ | $84.84_{\pm2.0}$ | $45.69_{\pm0.58}$ | $94.25_{\pm0.51}$ | $91.90_{\pm0.11}$ |
| | CLS-ER | $\mathbf{95.73}_{\pm0.11}$ | $\mathbf{86.78}_{\pm0.17}$ | $\mathbf{46.74}_{\pm0.31}$ | $94.25_{\pm0.06}$ | $\mathbf{92.03}_{\pm0.05}$ |

Table S2: Comparison of CLS-ER with Mean-ER (single semantic memory) on Class-IL and Domain-IL settings. We report the mean and 1 std of 10 runs with different initializations.

$\alpha_S, \alpha_P \in (0.99, 0.999)$, $r_S, r_P \in (0, 1]$, $\lambda \in (0, 2]$. Table S4 provides the hyperparameters used for each of the experimental settings. Note that for the vast majority of datasets, we use uniform settings (lr, epochs, batch size, memory batch size, and $lambda$) across the different buffer sizes and requires only slight modifications in the other hyperparameters which shows that our method does not require extensive finetuning for different memory budgets.

### E.1 GCIL-CIFAR-100

To test our method under challenging GIL settings that better simulate the challenges of CL in the real world, we incorporate the GCIL setting from the code provided by Mi et al. (2020) with the continual dataset template class in the mammoth framework. We set the number of phases (length of task sequences) to 20, with the total number of samples in each phase set to 1000 and the maximum number of classes in each phase set to 50. We evaluate on both uniform and longtail (imbalanced) data distributions. Since GCIL involves the probabilistic sampling of the classes and their samples in each phase, the random seed determines the complexity of the GCIL setting. Therefore, for reproduciblility and to gauge the stability of the methods, we fix the dataset seed to 1993 and report the average and standard deviation of 10 differently initialized models trained on the same settings.

For each of our method, we use identical training scheme (lr=0.1, epochs=100, batch size=32 and memory batch size=32). For DER++, as per the authors suggestion, we performed hyperparameter search over $\alpha \in [0.2, 0.3]$ and $beta \in [0.5, 1.0]$ with step size of 0.1. Table S5 provides the parameters chosen for each of the method under the different settings.

### E.2 PERTURBATION ANALYSIS

For the perturbation analysis, we used the code and checkpoints provided by Buzzega et al. (2020a) for DER++ and ER. We would like to express our gratitude to the authors for their support and for making the mammoth framework available for the research community which provides a framework for a fair comparison of different CL methods under uniform experimental conditions.

| $\lambda$ | $r_S$ | $r_P$ | Stable Model | Working Model | Plastic Model |
|---|---|---|---|---|---|
| 0.1 | 0.1 | 0.2 | $73.53_{\pm1.07}$ | $62.80_{\pm0.63}$ | $71.11_{\pm2.21}$ |
| | | 0.3 | $72.44_{\pm1.37}$ | $63.53_{\pm1.98}$ | $70.97_{\pm1.71}$ |
| | | 0.4 | $73.05_{\pm0.93}$ | $61.81_{\pm1.92}$ | $68.75_{\pm2.20}$ |
| | | 0.5 | $75.16_{\pm1.09}$ | $63.95_{\pm1.92}$ | $70.42_{\pm1.09}$ |
| | | 0.6 | $75.04_{\pm0.66}$ | $62.82_{\pm0.70}$ | $69.61_{\pm0.31}$ |
| | | 0.7 | $73.94_{\pm0.48}$ | $63.34_{\pm0.46}$ | $70.30_{\pm1.68}$ |
| | | 0.8 | $74.61_{\pm1.10}$ | $62.68_{\pm0.65}$ | $70.74_{\pm0.39}$ |
| | | 0.9 | $73.74_{\pm2.14}$ | $62.69_{\pm1.97}$ | $69.52_{\pm0.79}$ |
| | | 1.0 | $75.73_{\pm0.68}$ | $64.21_{\pm1.11}$ | $72.00_{\pm0.56}$ |
| | 0.2 | 0.3 | $70.26_{\pm1.79}$ | $61.63_{\pm1.03}$ | $69.31_{\pm1.82}$ |
| | | 0.4 | $71.80_{\pm1.17}$ | $62.64_{\pm0.18}$ | $70.64_{\pm1.22}$ |
| | | 0.5 | $70.69_{\pm2.13}$ | $61.76_{\pm0.64}$ | $69.65_{\pm1.92}$ |
| | | 0.6 | $72.45_{\pm0.68}$ | $63.87_{\pm0.85}$ | $71.29_{\pm0.72}$ |
| | | 0.7 | $71.47_{\pm1.98}$ | $61.12_{\pm1.90}$ | $70.22_{\pm2.24}$ |
| | | 0.8 | $72.16_{\pm0.56}$ | $62.71_{\pm0.57}$ | $70.83_{\pm0.64}$ |
| | | 0.9 | $72.09_{\pm0.59}$ | $63.33_{\pm1.01}$ | $71.20_{\pm0.87}$ |
| | | 1.0 | $72.05_{\pm1.35}$ | $63.74_{\pm1.75}$ | $71.01_{\pm1.28}$ |
| | 0.3 | 0.4 | $68.46_{\pm1.48}$ | $60.96_{\pm1.62}$ | $68.31_{\pm1.40}$ |
| | | 0.5 | $70.05_{\pm2.54}$ | $63.06_{\pm1.26}$ | $69.90_{\pm2.57}$ |
| | | 0.6 | $69.57_{\pm1.07}$ | $61.25_{\pm1.96}$ | $69.36_{\pm1.06}$ |
| | | 0.7 | $68.99_{\pm2.34}$ | $61.61_{\pm2.17}$ | $68.81_{\pm2.27}$ |
| | | 0.8 | $71.21_{\pm0.48}$ | $63.08_{\pm0.82}$ | $70.99_{\pm0.57}$ |
| | | 0.9 | $71.26_{\pm1.47}$ | $62.33_{\pm0.64}$ | $71.03_{\pm1.56}$ |
| | | 1 | $69.00_{\pm0.41}$ | $61.38_{\pm0.92}$ | $68.69_{\pm0.31}$ |
| 0.15 | 0.1 | 0.2 | $70.19_{\pm1.97}$ | $61.39_{\pm2.06}$ | $69.81_{\pm1.60}$ |
| | | 0.3 | $73.72_{\pm0.83}$ | $62.07_{\pm0.84}$ | $70.18_{\pm0.09}$ |
| | | 0.4 | $71.60_{\pm2.30}$ | $61.11_{\pm2.00}$ | $69.15_{\pm1.08}$ |
| | | 0.5 | $74.18_{\pm0.37}$ | $63.32_{\pm0.98}$ | $71.08_{\pm2.04}$ |
| | | 0.6 | $74.90_{\pm0.40}$ | $62.35_{\pm2.31}$ | $71.58_{\pm0.79}$ |
| | | 0.7 | $74.52_{\pm1.10}$ | $62.59_{\pm2.64}$ | $70.90_{\pm2.30}$ |
| | | 0.8 | $75.27_{\pm1.21}$ | $62.00_{\pm1.98}$ | $71.27_{\pm1.64}$ |
| | | 0.9 | $74.61_{\pm0.91}$ | $63.47_{\pm1.60}$ | $70.49_{\pm0.95}$ |
| | | 1.0 | $76.03_{\pm0.64}$ | $63.63_{\pm1.01}$ | $71.42_{\pm1.11}$ |
| | 0.2 | 0.3 | $72.59_{\pm1.44}$ | $61.81_{\pm1.03}$ | $72.02_{\pm1.30}$ |
| | | 0.4 | $71.30_{\pm3.42}$ | $63.15_{\pm0.51}$ | $70.92_{\pm2.83}$ |
| | | 0.5 | $69.89_{\pm1.95}$ | $60.60_{\pm0.95}$ | $68.87_{\pm2.56}$ |
| | | 0.6 | $72.34_{\pm0.89}$ | $62.18_{\pm1.31}$ | $71.15_{\pm0.94}$ |
| | | 0.7 | $72.70_{\pm1.11}$ | $62.50_{\pm1.18}$ | $71.49_{\pm1.21}$ |
| | | 0.8 | $72.42_{\pm1.50}$ | $61.85_{\pm0.83}$ | $71.04_{\pm1.68}$ |
| | | 0.9 | $71.18_{\pm0.71}$ | $61.81_{\pm1.29}$ | $70.09_{\pm0.54}$ |
| | | 1.0 | $73.52_{\pm0.65}$ | $64.19_{\pm0.86}$ | $72.56_{\pm0.52}$ |
| | 0.3 | 0.4 | $70.32_{\pm1.39}$ | $62.39_{\pm2.00}$ | $70.13_{\pm1.33}$ |
| | | 0.5 | $71.60_{\pm1.53}$ | $62.67_{\pm2.08}$ | $71.40_{\pm1.54}$ |
| | | 0.6 | $70.36_{\pm1.82}$ | $62.28_{\pm2.28}$ | $70.13_{\pm2.03}$ |
| | | 0.7 | $69.79_{\pm1.93}$ | $61.13_{\pm1.37}$ | $69.65_{\pm1.82}$ |
| | | 0.8 | $69.85_{\pm0.95}$ | $60.69_{\pm1.63}$ | $69.78_{\pm0.60}$ |
| | | 0.9 | $71.32_{\pm1.68}$ | $61.79_{\pm1.41}$ | $71.03_{\pm1.61}$ |
| | | 1.0 | $71.39_{\pm0.49}$ | $62.35_{\pm0.88}$ | $71.11_{\pm0.55}$ |

Table S3: The effect of different hyperparameter settings on the individual components of CLS-ER trained on S-CIFAR-10 with 500 buffer size. For all the experiments $\alpha_S$ and $\alpha_P$ are fixed to 0.999 and the performance is averaged over 3 runs with different initialization.

| Dataset | Buffer | lr | Epochs | Batch Size | Memory Batch Size | $\lambda$ | $\alpha_S$ | $\alpha_P$ | $r_S$ | $r_P$ |
|---|---|---|---|---|---|---|---|---|---|---|
| S-MNIST | 200 | 0.03 | 1 | 10 | 128 | 2.0 | 0.99 | 0.99 | 0.9 | 1.0 |
| | 500 | 0.1 | 1 | 10 | 32 | 2.0 | 0.99 | 0.99 | 0.9 | 1.0 |
| | 5120 | 0.1 | 1 | 10 | 32 | 2.0 | 0.99 | 0.99 | 0.8 | 1.0 |
| S-CIFAR-10 | 200 | 0.1 | 50 | 32 | 32 | 0.15 | 0.999 | 0.999 | 0.1 | 0.3 |
| | 500 | 0.1 | 50 | 32 | 32 | 0.15 | 0.999 | 0.999 | 0.1 | 0.9 |
| | 5120 | 0.1 | 50 | 32 | 32 | 0.15 | 0.999 | 0.999 | 0.8 | 1.0 |
| S-Tiny-ImageNet | 200 | 0.05 | 50 | 32 | 32 | 0.1 | 0.999 | 0.999 | 0.04 | 0.08 |
| | 500 | 0.05 | 50 | 32 | 32 | 0.1 | 0.999 | 0.999 | 0.05 | 0.08 |
| | 5120 | 0.05 | 50 | 32 | 32 | 0.1 | 0.999 | 0.999 | 0.07 | 0.08 |
| R-MNIST | 200 | 0.2 | 1 | 128 | 128 | 0.75 | 0.999 | 0.99 | 1.0 | 1.0 |
| | 500 | 0.2 | 1 | 128 | 128 | 0.75 | 0.999 | 0.99 | 1.0 | 1.0 |
| | 5120 | 0.2 | 1 | 128 | 128 | 0.75 | 0.999 | 0.99 | 1.0 | 1.0 |
| P-MNIST | 200 | 0.2 | 1 | 128 | 128 | 1.0 | 0.99 | 0.99 | 0.8 | 1.0 |
| | 500 | 0.2 | 1 | 128 | 128 | 1.0 | 0.99 | 0.99 | 0.8 | 1.0 |
| | 5120 | 0.2 | 1 | 128 | 128 | 1.0 | 0.99 | 0.99 | 0.9 | 1.0 |
| MNIST-360 | 200 | 0.2 | 1 | 16 | 16 | 0.75 | 0.999 | 0.99 | 1.0 | 1.0 |
| | 500 | 0.2 | 1 | 16 | 32 | 1.25 | 0.99 | 0.99 | 0.9 | 1.0 |
| | 1000 | 0.2 | 1 | 16 | 128 | 0.75 | 0.99 | 0.99 | 0.9 | 1.0 |
| GCIL-CIFAR-100 | 200 | 0.1 | 100 | 32 | 32 | 0.1 | 0.999 | 0.999 | 0.6 | 0.7 |
| | 500 | 0.1 | 100 | 32 | 32 | 0.1 | 0.999 | 0.999 | 0.6 | 0.7 |
| | 1000 | 0.1 | 100 | 32 | 32 | 0.1 | 0.999 | 0.999 | 0.6 | 0.8 |

Table S4: The hyperparameters used for each of the experimental settings for CLS-ER.

| Distribution | Buffer | lr | Epochs | Batch Size | Memory Batch Size | $\alpha$ | $\beta$ |
|---|---|---|---|---|---|---|---|
| Uniform | 200 | 0.1 | 100 | 32 | 32 | 0.2 | 0.5 |
| | 500 | 0.1 | 100 | 32 | 32 | 0.2 | 0.6 |
| | 1000 | 0.1 | 100 | 32 | 32 | 0.3 | 0.6 |
| Longtail | 200 | 0.1 | 100 | 32 | 32 | 0.2 | 0.6 |
| | 500 | 0.1 | 100 | 32 | 32 | 0.2 | 0.8 |
| | 1000 | 0.1 | 100 | 32 | 32 | 0.3 | 0.9 |

Table S5: The hyperparameters used for DER++ on GCIL-CIFAR-100 experiments. CLS-ER uses the same hyperparameters for both Uniform and Longtail settings (Table S4).

| Dataset | Buffer | lr | Epochs | Batch Size | Memory Batch Size | $\lambda$ | $\alpha$ | $r$ |
|---|---|---|---|---|---|---|---|---|
| S-MNIST | 200 | 0.03 | 1 | 10 | 128 | 2.0 | 0.99 | 1.0 |
| | 500 | 0.1 | 1 | 10 | 32 | 2.0 | 0.99 | 1.0 |
| | 5120 | 0.1 | 1 | 10 | 32 | 2.0 | 0.99 | 1.0 |
| S-CIFAR-10 | 200 | 0.1 | 50 | 32 | 32 | 0.15 | 0.999 | 0.2 |
| | 500 | 0.1 | 50 | 32 | 32 | 0.15 | 0.999 | 0.5 |
| | 5120 | 0.1 | 50 | 32 | 32 | 0.15 | 0.999 | 0.8 |
| S-Tiny-ImageNet | 200 | 0.05 | 50 | 32 | 32 | 0.1 | 0.999 | 0.06 |
| | 500 | 0.05 | 50 | 32 | 32 | 0.1 | 0.999 | 0.08 |
| | 5120 | 0.05 | 50 | 32 | 32 | 0.1 | 0.999 | 0.08 |
| R-MNIST | 200 | 0.2 | 1 | 128 | 128 | 0.75 | 0.999 | 1.0 |
| | 500 | 0.2 | 1 | 128 | 128 | 0.75 | 0.999 | 1.0 |
| | 5120 | 0.2 | 1 | 128 | 128 | 0.75 | 0.999 | 1.0 |
| P-MNIST | 200 | 0.2 | 1 | 128 | 128 | 1.0 | 0.99 | 0.9 |
| | 500 | 0.2 | 1 | 128 | 128 | 1.0 | 0.99 | 1.0 |
| | 5120 | 0.2 | 1 | 128 | 128 | 1.0 | 0.99 | 0.9 |

Table S6: The hyperparameters used for each of the experimental settings for Mean-ER.

