# OpenReview forum: "Learning Fast, Learning Slow: A General Continual Learning Method based on Complementary Learning System"
_ICLR.cc/2022/Conference — ICLR 2022 Poster_

### Official Review · Reviewer_M4Au · 2021-10-31

**Correctness:** 3
**Technical Novelty And Significance:** 3
**Empirical Novelty And Significance:** 3
**Recommendation:** 6
**Confidence:** 4

**Main Review:**

Strengths:

1. The proposed method is well motivated and novel in the context of continual learning. Maintaining multiple models for different purposes is interesting and is shown to be helpful for continual learning.

2. The writing and organization is clear. It is easy to understand the proposed algorithm and the details.

3. Different continual learning settings are considered, the proposed approach shows better results across all the settings. The analysis of different models also make it easy to understand how the method works.


Weaknesses:

1. One of the main drawbacks of the proposed approach is that three models need to be maintained during training which incur a heavy cost.   One question is that if reducing the individual model size while keeping the whole model size the same, what's the performance in this case?

2. The proposed method has a lot of hyperparameters to tune, which may be a problem if the method is applied to a different task.


More questions:

1. Why update the plastic and stable models stochastically?

2. In table 1, why CLS-ER is better than JOINT on S-MNIST? JOINT should be the upper bound performance?

3. The authors make a connection to flat minima, if explicitly enforcing flat minima, would the proposed method can still do better?

**Summary Of The Paper:**

In this paper, motivated from the CLS theory in neuroscience that efficient learning requires short-term adaptation and slow learning of structured information, the authors propose a novel dual memory experience replay method for continual learning. The idea is to build  long-term and short-term semantic memories by maintaining another two models: plastic model and stable model. The plastic model is used for fast learning of recent experiences and the stable model is used for slow learning of structural knowledge. The two models are updated with a Mean Teacher fashion during training with different frequencies. The proposed CLS-ER is  task-agnostic and can be applied in different continual learning settings. The experimental results show that the proposed CLS-ER outperforms several baselines for continual learning.

**Summary Of The Review:**

In this paper, a novel idea of continual learning is proposed. The authors clearly describe the proposed approach and also give a detailed analysis. Overall, it is a well-written paper with an interesting idea. The results also show the benefits of the proposed approach.

---

> ### Author Response · Authors · 2021-11-20
> **Response to Reviewer M4Au**
>
> Thank you for your encouraging remarks and valuable suggestions. We are glad you found it interesting and hope to address your queries below.
>
> > **One of the main drawbacks of the proposed approach is that three models need to be maintained during training which incur a heavy cost. One question is that if reducing the individual model size while keeping the whole model size the same, what's the performance in this case?**
>
> We acknowledge that the memory complexity of our method is higher because of the additional cost of maintaining the semantic memories. Thank you for the valuable suggestion, we will explore this idea in addition to more efficient ways of modeling the semantic memories. (Please see the Q1 response to Reviewer MUyX).
>
> > **The proposed method has a lot of hyperparameters to tune, which may be a problem if the method is applied to a different task.**
>
> This is a valid concern and while our method does have a number of hyperparameters, we believe that the complementary nature of the individual components in our method and its ability to attain similar performance for different choices of hyperparameters,  reduces the required fine-tuning effort considerably. Specifically, Section B.5 shows that it is not highly sensitive to the particular choice of hyperparameters as different settings can attain similar performance. Also, because of the complementary nature of the components, we can keep the majority of the parameters fixed (e.g. $\lambda$, $\alpha_S$, $\alpha_S$ and $r_S$) and only finetune the remaining parameters (e.g. $r_P$) which facilitates hyperparameter tuning significantly. Table S.4 further shows that for the vast majority of our experiments we use the same parameters for different buffer sizes.
>
> > **Why update the plastic and stable models stochastically?**
>
> Thank you for bringing this point, we will provide detail in the revision. Updating the semantic memories stochastically compared to a deterministic approach (e.g updating the plastic model every 3rd epoch and stable model every 6th iteration) reduces the overlap in the snapshots of the working model and leads to more diversity in semantic memories and hence higher complementary knowledge. Interesting, noise is a salient component of the computational strategy of the brain [1] and permeates every level of the nervous system [2] which makes stochastic updates more biologically plausible. Empirically we also found that stochastic update performs better than fixed periodic update.
>
> > **In table 1, why CLS-ER is better than JOINT on S-MNIST? JOINT should be the upper bound performance?**
>
> This is a good point and it is indeed the case that JOINT acts as the upper bound and our method achieves performance close to JOINT only for the simpler MNIST settings. For S-MNIST, while the average for CLS-ER is only slightly higher, it is within 1 standard deviation of the JOINT performance. We also conjecture that perhaps because of the relative simplicity of the task, with a higher buffer size (i.e. 5120), there might be sufficient samples to learn a good classifier and our method provides better regularization to avoid overfitting and hence leads to slightly higher generalization.
>
> > **The authors make a connection to flat minima, if explicitly enforcing flat minima, would the proposed method can still do better?**
>
> Thank you for the suggestion. We are not aware of any method that explicitly enforces flatter minima besides perhaps higher entropy regularization[3]. We believe that additional regularizations on top of our method can be an interesting avenue for future research.
>
> Please let us know if we failed to address your concerns or if you require further information or explanation.
>
> **References:**
>
> [1] Maass, Wolfgang. "Noise as a resource for computation and learning in networks of spiking neurons." Proceedings of the IEEE 102.5 (2014): 860-880.
>
> [2] Faisal, A. Aldo, Luc PJ Selen, and Daniel M. Wolpert. "Noise in the nervous system." Nature reviews neuroscience 9.4 (2008): 292-303.
>
> [3] Chaudhari, Pratik, et al. "Entropy-sgd: Biasing gradient descent into wide valleys." Journal of Statistical Mechanics: Theory and Experiment 2019.12 (2019): 124018.

---

### Official Review · Reviewer_MUyX · 2021-11-01

**Correctness:** 3
**Technical Novelty And Significance:** 2
**Empirical Novelty And Significance:** 2
**Recommendation:** 5
**Confidence:** 2

**Main Review:**

The method is very simple to understand and mimicking the CLS theory with the hippocampus and neocortex is interesting.

Compared to existing methods, is it a fair comparison in terms of training cost such as using two models including plastic and stable models and using reply data to train the working model?

I think more analysis about plastic and stable models are needed because these model act like hippocampus and neocortex based on CLS theory. What is optimal semantic? Just select the logit with the highest softmax between the two models?

**Summary Of The Paper:**

They propose a dual memory experience based on complementary learning systems (CLS).
The working model is updated using consistency loss with the selected optimal semantic memory from plastic and stable models.
Plastic and stable models are updated with an exponential moving average trick (EMA) and the working model.

**Summary Of The Review:**

I am not an expert in this domain and have not much experience with related works. However, the idea is very simple and reasonable and the results outperform previous existing methods. However, more analysis is needed for the claim.

---

> ### Author Response · Authors · 2021-11-20
> **Response to Reviewer MUyX**
>
> Thank you for your feedback. We are glad that you found the idea of incorporating the hippocampus and neocortex into DNNs interesting as it was the main goal of our study to show the potential of mimicking the complementary learning system for enabling efficient continual learning. We hope to address your concerns below.
>
>
> > **Q1: Compared to existing methods, is it a fair comparison in terms of training cost such as using two models including plastic and stable models, and using reply data to train the working model?**
>
> We acknowledge that the memory complexity of our method is higher because of the additional cost of maintaining the semantic memories. This is, however, still more efficient compared to some knowledge distillation or a mixture of experts approaches that maintain a separate network for each task [1] or dynamic neural networks where the network parameters grow with the tasks. Our method maintains a constant size and can scale to a large number of tasks. Also, we would like to emphasize that while the training cost is higher, our method does not require any additional computations for inference as only the stable model is used, requiring a single forward pass similar to the other methods. Our goal was to bridge the gap between DNN and the brain which utilizes multiple memory systems and show the potential benefits of incorporating such a dual memory architecture. In future work, we aim to further explore the design space for brain-inspired algorithms for mimicking the hippocampus and neocortex considering both the faithfulness to the brain and the energy efficiency.
>
> > **Q2: I think more analysis about plastic and stable models are needed because these model act like hippocampus and neocortex based on CLS theory. What is optimal semantic? Just select the logit with the highest softmax between the two models?**
>
> We apologize for the confusion and will provide more intuition in the revision. The plastic and stable model aim to mimic the rapid and slow adaptation of information respectively which we aim to achieve through the design of exponential moving averaged models with different update frequencies and window sizes. Section B.2 and Figures S1 and S2 in the supplementary, which show the task-wise performance of each of the models as training progresses, demonstrate empirically that our approach does indeed achieve the intended purpose.
>
> Optimal semantic information refers to the consolidated structural knowledge encoded in the model logits for the efficient replay which account for the consolidation of feature space and adaptation of the decision boundaries of the previous tasks compared to the sub-optimal local logits in DER++. We highlight the benefits of using these consolidated logits instead of the local logits in DER++ in our results section and provide intuition for how they add value in each of the evaluation settings.
>
> We aim to extract the optimal consolidated logits for replay for a memory sample, which by design would come from the stable model for a memory sample from an earlier task and the plastic model for a recent task (as can be seen in the task-wise performance of the semantic memories). In line with our goal to provide a general incremental learning method, we avoid using the task boundaries to guide our selection and instead use the simple approach of utilizing the performance of the semantic memories as the selection criterion which works well in practice as our method is able to achieve the intended distribution of performance with the stable and plastic model. Further research into the selection criterion is an interesting research avenue for future work.
>
> We hope to have addressed your concerns and would be happy to provide more details.

---

### Official Review · Reviewer_azqK · 2021-11-01

**Correctness:** 4
**Technical Novelty And Significance:** 2
**Empirical Novelty And Significance:** 3
**Recommendation:** 8
**Confidence:** 3

**Main Review:**

I enjoyed reading this publication. I'm not very familiar with the latest CL methods but the authors address various related works and evaluate all models thoroughly. I think the setting without task boundaries is harder and more interesting since the real world also doesn't come with clear task boundaries. The paper is following recent work towards robust and extensive evaluation which is more difficult but will help the CL community to make real progress.

The empirical upper bound "JOINT" is very instructive and highlights that there is still room for progress. That said, the JOIN performance seems low given the excellent performance of other models on MNIST, CIFAR, Imagenet datasets. Is this because the models are very small? Does CLS-ER work also with larger models?

The flat mima discussion is less essential but is probably missing a reference to Flat Minima, Hochreiter et al 1997.

Instead of the interesting section 6 I'd have preferred some ablation experiments (though the component analysis is appreciated). E.g. are two teacher networks necessary? How much worse is the model if one teacher network is used (with a tuned rate)? What if there were three networks (each with their own rate)?

I did not understand how samples are removed from the buffer (the episodic memory). Are samples removed eventually? If yes, how and where is that explained? If not, what is the space complexity of algorithm 1?

**Summary Of The Paper:**

The paper presents CLS-ER, a dual memory mechanism for the continual learning setting. During training the model receives data from a non-i.i.d. source as well as random samples from a sample buffer (the episodic memory). The stable and plastic models act as teacher models. Their logits are the target of the student model. Which one is used as a target depends on the highest softmax score w.r.t. a ground trough class. The teacher networks are both exponential moving averages of the previous student models. Thus, the only difference between the teacher models "plastic" and "stable" is their EMA rates which in the case of the plastic model is larger such that it focuses on more recent weight updates.  The slower learning of the stable model is the mechanism which ought to mirror the consolidation of structural knowledge. It is used from inference Samples from the datastream are transferred to the sample buffer using reservoir sampling.

The authors experiment on image datasets (variants of MNIST, CIFAR10, Tiny-ImageNet, see the Mammoth framework) with data augmentation (random flips and crops). They compare with several prio works and the evaluation is thorough and convincing.

**Summary Of The Review:**

I think this work is good in general. The writing is great, the evaluation seems thorough, and comparisons look fair. The method is easy to understand and the empirical results are convincing to me. The authors discuss recent and related works and their method seems original but I lack the expertise to judge the novelty of the approach. It builds significantly on previous work (DER) so I'd argue that some aspects of the publication exist in previous work.

---

> ### Author Response · Authors · 2021-11-20
> **Response to Reviewer azqK**
>
> Thank you for your time and valuable feedback. It is quite encouraging that you enjoyed reading our work and found the evaluation convincing. We hope to address your concerns below.
>
> > **the JOIN performance seems low given the excellent performance of other models on MNIST, CIFAR, Imagenet datasets. Is this because the models are very small? Does CLS-ER work also with larger models?**
>
> Thank you for pointing this out, it would be great though if you could clarify a bit on the reference models so that we could provide a more specific response as there are several factors that can affect the model performance, including model size, training scheme, data augmentations, and regularization techniques.
>
> For a fair comparison, we keep uniform training settings so that we can disentangle the effect of the continual learning method from these other factors. We follow the experimental settings (model and training scheme) from [1] and provide the baseline results from their study.
> Following them, we use a small fully connected network for MNIST based settings as it has sufficient capacity given the data complexity and a ResNet-18 model for CIFAR and TinyImageNet settings which is a common choice for in literature, as well as for continual learning methods.
> We have not tried our method for models larger than ResNet-18, but we see no reason for the method not to scale up to larger models. In fact, we would argue that larger models would provide a higher capacity for learning a large number of tasks and richer representations in the semantic memories. Please let us know if we should provide more details.
>
> > **The flat minima discussion is less essential but is probably missing a reference to Flat Minima, Hochreiter et al 1997.**
>
> Thank you for sharing the citation. We will add it in the revision.
>
> > **Instead of the interesting section 6 I'd have preferred some ablation experiments (though the component analysis is appreciated). E.g. are two teacher networks necessary? How much worse is the model if one teacher network is used (with a tuned rate)? What if there were three networks (each with their own rate)?**
>
> This is a good suggestion that would provide valuable insights. We opted for two semantic memories specifically as our study aims to mimic the fast and slow learning mechanisms in the hippocampus and neocortex. We had evaluated the method using a single teacher model and found it to perform sub-par with the dual memory counterpart as one model fails to keep the performance on both the recent and earlier tasks together i.e there is an inherent trade-off. Tuning the semantic memory to adapt to the recent changes comes at the cost of performance on earlier tasks and vice versa. We will add these results in the supplementary.
>
> We did not try our method for more than two semantic memories (teachers) as we did not have a good intuition for the role of additional semantic memories. However, our method is flexible enough to incorporate more than two teachers, and exploring their effect could be interesting.
>
> > **I did not understand how samples are removed from the buffer (the episodic memory). Are samples removed eventually? If yes, how and where is that explained? If not, what is the space complexity of algorithm 1?**
>
> Sorry for the lack of clarity on this, we will add more details of reservoir sampling in the revision as well as add the algorithm in the supplementary. The samples are indeed removed from the buffer. A fixed-size memory buffer is maintained using reservoir sampling which assigns equal probability to each sample in the stream for being represented in the buffer. Accordingly, each sample in the data stream has an equal probability of being added to the memory buffer by replacing the existing samples in the buffer at random. Reservoir sampling does not assign any priority to the samples being added or replaced from the buffer.
>
> Please let us know if we failed to address any of your concerns and if we can provide any additional information.

---

> > ### Comment · Reviewer_azqK · 2021-11-25
> > **Response**
> >
> > Thank you for your response. After reading the other reviews and your responses I still recommend acceptance. For future work, I'd recommend to "upgrade" the model size, training scheme, etc to better performing models. That said, I see no reason why the contributions of this paper would not apply there.
> >
> > One thing that bothers me is that your decision to use two models is solely based on CLS theory. I think it is fine to use neuroscience as an inspirational source but these models are far far far removed from simulations of real neurological systems like the hippocampus. Furthermore, the goal is not to simulate such systems but to perform well on the classification/regression problem. Thus, _all_ your central design decisions ought to be based on (mathematical) theory or empirical data (either yours or others). In the context of this paper and this conference, "because CLS theory says so" is in my opinion not enough reason to do things a certain way. That said, your results are convincing and I have otherwise no critique, thus my recommendation to accept.

---

### Official Review · Reviewer_8Zhu · 2021-11-02

**Correctness:** 4
**Technical Novelty And Significance:** 4
**Empirical Novelty And Significance:** 4
**Recommendation:** 6
**Confidence:** 5

**Main Review:**


Strengths:
1, Semantic memory is the first leveraged in incremental learning, which is interesting in my opinion. The contribution of this paper is novel.
2, The codes are released, which guarantees the reproduction of the proposed method.
3, The experiment results prove the effectiveness of the proposed method, which obtains impressive performance.

Weaknesses:
1, Can this method be applied to online continual learning task?
2, The experiments for domain-IL are only evaluated in one dataset. Experiment results on more datasets should be shown in this paper.
3, For Class-IL, some experiments with different base classes should be shown in this paper.


**Summary Of The Paper:**

This paper proposes a novel dual memory experience replay method to store the knowledge of previous tasks. In addition, long-term and short-term semantic memories are leveraged to replay the neural activities of the episodic memories and align the decision boundary.

**Summary Of The Review:**

In my opinion, the contribution of the proposed method is novel and interesting. In addition, the writing of this paper is clear and easy to understand. The experiments are sufficient to prove the effectiveness of the proposed methods. However, more experiments should be designed to prove the proposed method further.

---

> ### Author Response · Authors · 2021-11-20
> **Response to Reviewer 8Zhu**
>
> Thank you for your kind words and constructive feedback. We are glad that you found the method interesting and hope to address your questions below.
>
> > **Can this method be applied to online continual learning tasks?**
>
> This is an interesting question and actually a research direction that we aim to extend our method to in the future. We would also like to emphasize that the experiments on MNIST based settings (S-MNIST, R-MNIST, P-MNIST, and MNIST-360) can be considered under the online continual learning scenario as we only train the network for 1 epoch, and thereby the model only sees the data for each task once as the common single epoch protocol in online continual learning. Our results on these settings demonstrate the potential of CLS-ER as an efficient online continual learning method.
>
> Furthermore, as highlighted by [1] in their supplementary section F.3, we also believe that it is necessary to set the number of epochs per task considering the dataset complexity in order to disentangle the effects of catastrophic forgetting from those of underfitting. We share their suggestion that future CL works should strive for realism by designing experimental settings which are in line with the guidelines of General Continual Learning[2] which is the goal of our study rather than adopting the single-epoch protocol.
> As such, we do not see any limitation of our method that would impede its application in more challenging online continual learning settings.  Thank you for bringing this point, we will highlight it more in the revision.
>
> > **The experiments for domain-IL are only evaluated in one dataset. Experiment results on more datasets should be shown in this paper.**
>
> We are sorry that you got this impression, we will make it clearer in the revision. We had also evaluated our method on Permuted-MNIST which together with Rotated MNIST are the two most common benchmark settings for domain-IL. We had included the results in the supplementary section B.1 and our method is effective under this setting as well. We had excluded it from our main paper as, in accordance with [2], we do not consider it as a viable evaluation protocol as it violates the cross-task resemblance desiderata and deviates from the goals of continual learning.
>
> Furthermore, the MNIST-360 setting can, in part, be considered as evaluating the method on domain-IL as it exposes the model to a smooth rotational distribution shift in addition to a sharp distribution shift (details in supplementary section A.3). We believe that the performance of our method on these three settings collectively demonstrate the effectiveness of CLS-ER for the domain-IL setting. Thank you for bringing this to our attention. For completion, we will add the results of Permuted-MNIST in the main table and highlight the domain-IL learning aspect of MNIST-360 more.
>
> > **For Class-IL, some experiments with different base classes should be shown in this paper.**
>
> We are not sure if we understood the comment correctly, please let us know if our response fails to address your concern, we would be happy to provide more details.
> For the Class-IL setting, we follow the protocol in literature whereby we start with a randomly initialized model (without any pre-training) and train them on a sequence of tasks with a disjoint set of classes. This is in contrast to the common approach in many online continual learning settings, where the model is initially trained on a large set of base classes.
>
> We hope to have addressed your concerns and would incorporate your suggestions in the revision. Please let us know if we can provide more details or explanations.
>
>
> **References:**
>
> [1] Buzzega, Pietro, et al. "Dark Experience for General Continual Learning: a Strong, Simple Baseline." 34th Conference on Neural Information Processing Systems (NeurIPS 2020). 2020.
>
> [2] Farquhar, Sebastian, and Yarin Gal. "Towards robust evaluations of continual learning." arXiv preprint arXiv:1805.09733 (2018).

---

### Author Response · Authors · 2021-11-22
**Revision Uploaded**

We thank the reviewers for their valuable feedback and suggestions. We have incorporated their suggestions into the revised submission and made the following changes:

- We performed additional ablation experiments to compare the performance of our method with a single semantic memory (Section D).
- Added the Permuted-MNIST results to the main text (Table 1).
- Highlighted the application of our method on online continual learning setting in the main text (and more details in Section A.4).
- Elaborated the details of reservoir sampling in the main text and added the algorithm in the appendix (Section B).
- Explained the reason for stochastic update of semantic memories (Section 3.3, 4th Paragraph).
- Provided clarification on optimal semantic memories (Section 3.3. 2nd Paragraph)
- Added missing citation for Flat Minima.

We hope that the revision and individual responses addressed the concerns of the reviewers. We would be happy to address any further concerns or questions.

---

### Public Comment · ~Quang_Pham1 · 2022-02-04
**References Suggestion**

Dear Authors,

Congratulations on the accepted paper at ICLR 2022. We found your paper interesting and shared some similarities with our earlier works of building a fast and slow learning continual learning system based on the CLS theory [1,2]. For example, in our DualNet[2], we also separated the system into fast and slow learners and proposed to train the slow learner by the Look-ahead optimizer, which is similar to your approach of using an exponential moving average of the working model.

We are aware that the NeurIPS 2021 decision and ICLR 2022 submission deadline were quite close, and you guys might have missed our earlier works during the submission. We would be appreciated it if you could include references to acknowledge our earlier papers in your camera-ready version. We believe that our works might have been developed concurrently, and adding references will further strengthen your paper and make it more comprehensive.

Best regards,

Quang

[1] Pham, Quang, et al. "Contextual transformation networks for online continual learning." International Conference on Learning Representations. 2021.

[2] Pham, Quang, Chenghao Liu, and Steven Hoi. "DualNet: Continual Learning, Fast and Slow." Advances in Neural Information Processing Systems 34 (2021).

---

> ### Public Comment · ~Fahad_Sarfraz1 · 2022-02-07
> **Author's Response**
>
> Congratulations on your paper being accepted at NeurIPS. It is good to see that more research work is being inspired by the complementary learning systems theory as we consider it as a promising direction for enabling efficient CL in DNNs. Thank you for bringing your paper to our attention, it seems to have been made public after our submission and presents a different approach than ours to incorporate a fast and slow learning mechanism with different learning objectives and interaction between the two systems. Our work focuses on maintaining long-term and short-term semantic memories and their interaction with the working memory for effective replay. We believe that both our works provide a compelling case for exploring the design space of multiple memory systems inspired by CLS theory in the brain. We hope our works would inspire the research community at large to work in this promising direction.

---

### Decision · Program_Chairs · 2022-01-20

**Decision:**

Accept (Poster)

**Comment:**

The manuscript proposes an experience replay method that supports two time scales of memory, as in complementary learning systems from the cognitive sciences literature. The authors demonstrate that their method on a wide range of benchmarks and after the rebuttal demonstrate it on one additional benchmark.
The reviewers raised a number of questions and concerns around additional experiments (benchmarks and ablations), online training, the cost of training, number of hyperparameters, further analysis, clarifications, and citations. The authors address the majority of these concerns during the rebuttal period, and overall the reviewers were in favour of acceptance. Therefore, I recommend acceptance of this work.